# Genetic analysis reveals functions of atypical polyubiquitin chains

**Fernando Meza Gutierrez[1†], Deniz Simsek[2†], Arda Mizrak[1], Adam Deutschbauer[3], Hannes Braberg[4], Jeffrey Johnson[4], Jiewei Xu[4], Michael Shales[4], Michelle Nguyen[5], Raquel Tamse-Kuehn[5], Curt Palm[5], Lars M Steinmetz[5], Nevan J Krogan[4], David P Toczyski[1]\***

[1]Department of Biochemistry and Biophysics, University of California, San Francisco, San Francisco, United States; [2]Amgen Research, San Francisco, United States; [3]Lawrence Berkeley National Laboratory, Berkeley, United States; [4]Department of Cellular and Molecular Pharmacology, University of California, San Francisco, San Francisco, United States; [5]Stanford Genome Technology Center, Stanford University, Stanford, United States

**Abstract** Although polyubiquitin chains linked through all lysines of ubiquitin exist, specific functions are well-established only for lysine-48 and lysine-63 linkages in *Saccharomyces cerevisiae*. To uncover pathways regulated by distinct linkages, genetic interactions between a gene deletion library and a panel of lysine-to-arginine ubiquitin mutants were systematically identified. The K11R mutant had strong genetic interactions with threonine biosynthetic genes. Consistently, we found that K11R mutants import threonine poorly. The K11R mutant also exhibited a strong genetic interaction with a subunit of the anaphase-promoting complex (APC), suggesting a role in cell cycle regulation. K11-linkages are important for vertebrate APC function, but this was not previously described in yeast. We show that the yeast APC also modifies substrates with K11-linkages in vitro, and that those chains contribute to normal APC-substrate turnover in vivo. This study reveals comprehensive genetic interactomes of polyubiquitin chains and characterizes the role of K11-chains in two biological pathways.

DOI: https://doi.org/10.7554/eLife.42955.001

\*For correspondence:
dpt4darwin@gmail.com

†These authors contributed equally to this work

## Introduction

The ubiquitination of proteins is a highly conserved posttranslational modification that can alter the stability, localization, and function of proteins (*Finley et al., 2012*). Ubiquitination regulates a broad range of important cellular pathways including the DNA damage response (*Jackson and Durocher, 2013*; *Harper and Elledge, 2007*), apoptosis (*Broemer and Meier, 2009*), cell cycle progression (*Borg and Dixit, 2017*), immune and inflammatory responses (*Wertz and Dixit, 2010*), and transcription (*Geng et al., 2012*). Three enzyme classes mediate the covalent attachment of ubiquitin, a 76 amino acid protein, to target proteins: An E1 ubiquitin-activating enzyme activates ubiquitin for transfer to an E2 ubiquitin-conjugating enzyme, which in turn interacts with an E3 ubiquitin ligase to transfer ubiquitin to lysine residues on the target protein (*Hershko and Ciechanover, 1998*). This enzymatic cascade can also transfer ubiquitin onto lysine residues of ubiquitin itself, thereby generating chains of polyubiquitin on target proteins (*Hershko and Ciechanover, 1998*).

Ubiquitin has seven acceptor lysines (K6, K11, K27 K29, K33, K48, and K63), all of which are used in vivo to generate polyubiquitin chains (*Xu et al., 2009*). Polyubiquitin can also be linked through the N-terminal amino group of methionine one in ubiquitin, termed linear ubiquitin chains, although to our knowledge, this has not been documented in *S. cerevisiae*. K48 and K11 are the most abundant ubiquitin linkage types—each accounting for about a third of ubiquitin linkages in yeast—while

the remaining five are present in relatively lower amounts (*Xu et al., 2009*). Ubiquitin chains vary in their three-dimensional conformations. K48-linked chains, for example, adopt a closed conformation with ubiquitin's hydrophobic patch being sequestered at the interface between two adjacent ubiquitin protomers (*Komander and Rape, 2012*). In contrast, K63-linked chains assume an extended conformation devoid of non-covalent contacts between ubiquitin monomers (*Datta et al., 2009*).

The structural diversity of polyubiquitin chains gives rise to disparate fates for protein targets (*Komander and Rape, 2012*). K48-linked chains, for example, specify proteins for proteasomal degradation (*Chau et al., 1989*), while K63-linked chains have non-degradative roles in various signaling pathways including the DNA damage response (*Harper and Elledge, 2007*; *Xu et al., 2009*; *Ulrich, 2005*; *Spence et al., 1995*; *Wang and Elledge, 2007*), protein trafficking (*Lauwers et al., 2009*; *MacGurn et al., 2012*), mitophagy (*Ordureau et al., 2015*), inflammation (*Deng et al., 2000*; *Grabbe et al., 2011*; *Wang et al., 2001*), and immune responses (*Manzanillo et al., 2013*). Degradative functions of K11 linkages in endoplasmic reticulum biology as well as Hedgehog signaling in fruit flies have been described (*Xu et al., 2009*; *Zhang et al., 2013*), while the critical role of K11-linkages in mitotic progression in metazoans has been studied in detail (*Meyer and Rape, 2014*; *Williamson et al., 2009*; *Matsumoto et al., 2010*; *Wu et al., 2010*; *Jin et al., 2008*; *Yau et al., 2017*). Although K6, K27, K29, and K33 ubiquitin linkages are especially rare in cells (*Xu et al., 2009*; *Dammer et al., 2011*), important observations have contributed to an increasing recognition of their physiological relevance (*Akutsu et al., 2016*). K6-linked polyubiquitin chains, for example, are reported to function in the DNA damage response through the BRCA1-BARD1 E3 ligase in a proteolysis-independent manner, as well as in mitophagy, through the E3 ligase, Parkin (*Ordureau et al., 2015*; *Nishikawa et al., 2004*; *Christensen et al., 2007*; *Ordureau et al., 2014*). K27-linkages have also been linked to mitophagy, as some Parkin substrates are reportedly decorated with K27-linked ubiquitin chains (*Birsa et al., 2014*; *Geisler et al., 2010*). mRNA stability is regulated by K29-linked ubiquitin chains on HuR, an mRNA binding protein, via an interaction with UBXD8, an adaptor for the essential p97 ATPase (*Zhou et al., 2013*). Finally, K33-linked ubiquitin chains have been suggested to regulate post-Golgi protein trafficking by mediating the interaction of Coronin-7 with Eps15, a *trans*-golgi network protein (*Yuan et al., 2014*).

Notwithstanding these and other intriguing observations (*Silva et al., 2015*; *Song et al., 2010*; *Elia et al., 2015*), a comprehensive examination of the roles of K6, K11, K27, K29, and K33 ubiquitin linkages is lacking, particularly in yeast. This is owed to both the partial redundancy of ubiquitin acceptor lysines (*Xu et al., 2009*), and the technical limitations of current protein analysis methods (*Swatek and Komander, 2016*; *Carrano and Bennett, 2013*). A high-throughput approach to uncover physiologically important functions of specific ubiquitin linkage types would significantly advance current understanding of the ubiquitin-proteasome system. Genetic interaction analysis, namely the study of the phenotypic effects of combining mutations, has long been a powerful tool for discovering novel gene-functions by revealing functional relationships between genes (*Dixon et al., 2009*; *Tong et al., 2001*; *Tong et al., 2004*; *Schuldiner et al., 2005*; *Braberg et al., 2013*; *Costanzo et al., 2010*; *Costanzo et al., 2016*; *Sarin et al., 2004*).

This report describes a comprehensive genetic analysis to uncover pathways that are regulated by specific ubiquitin linkage types. In a high-throughput manner, single deletions of non-essential genes were combined with mutations eliminating specific ubiquitin linkages. The growth phenotypes of double mutant strains were quantified to identify thousands of candidate genetic interactions. The genetic interactome of K11 linkages was of special interest due to that linkage type's relatively high abundance in yeast (*Xu et al., 2009*), and its known functions in the metazoan cell cycle (*Meyer and Rape, 2014*). Our genetic, molecular and biochemical analysis led us to the discovery of novel physiological functions of K11 linkages in promoting amino acid import and contributing to cell cycle progression. Our data are consistent with a model whereby the human and yeast APC both use K48 and K11 linkages to attain maximal efficiency, but they do so in a reciprocal fashion; in humans, K48 is part of a base chain from which homogenous K11-linked chains are extended, whereas in yeast, K11 is the critical linkage for a base chain, from which homogenous K48 chains are extended.

## Results

### The ubiquitin linkage synthetic genetic array

A synthetic genetic array analysis (SGA) was undertaken to identify pathways that are regulated by specific polyubiquitin chain types (*Tong et al., 2004*). Yeast strains constitutively expressing single, double, and triple lysine-to-arginine (K-to-R) mutant ubiquitin alleles were engineered by modifying all four loci from which ubiquitin is normally expressed in yeast (*Finley et al., 2012*) (*Figure 1A*, Materials and methods, *Figure 1—figure supplement 1*). The engineered yeast strains express ubiquitin at levels comparable to wild-type yeast (*Figure 1—figure supplement 2*). A strain expressing low levels of wild-type ubiquitin (lacking ubiquitin expression at the modified *ubi4* locus) was also included in the ubiquitin SGA to control for possible non-specific effects of perturbing ubiquitin levels (*Figure 1—figure supplement 2*, *Figure 1—figure supplement 3A*, Materials and methods). Because lysine 48 of ubiquitin is essential (*Finley et al., 1994*), strains expressing K48R ubiquitin also contained 20% wild-type ubiquitin (*Figure 1—figure supplement 3B*, Materials and methods). K63R ubiquitin mutants were initially included, but analysis of the data was not possible due to their extreme hypersensitivity to canavanine (data not shown), a toxic arginine analog used in the SGA protocol, even in the context of the deletion of the *CAN1* sensitivity gene (*Figure 1—figure supplement 4*). Linear ubiquitin chains were not analyzed in this study. The lysine-to-arginine ubiquitin mutant strains were systematically mated to a gene deletion library. The resulting diploid cells underwent sporulation to generate haploid double mutant cells expressing mutant ubiquitin alleles and carrying single gene deletions (*Figure 1—figure supplement 4*). Colony sizes of the approximately forty-five thousand pairwise combinations were measured to identify genetic interactions.

Previous SGA screens have examined the synthetic phenotype of two (*Tong et al., 2004*; *Schuldiner et al., 2005*; *Braberg et al., 2013*; *Costanzo et al., 2016*; *Tong and Boone, 2006*; *Collins et al., 2010*) or, in a few cases, three mutations (*Haber et al., 2013*), whereas the ubiquitin SGA necessitates examining five loci. During conventional SGA analysis in the S288C strain, heterozygous diploids are sporulated to generate haploid cells, a subset of which contains the double mutant to be examined. The haploid selection steps require the co-segregation of an additional three haploid selectable markers to efficiently eliminate unsporulated diploid cells that would otherwise generate background (*Collins et al., 2010*). The desired haploid spores are relatively abundant since only five total loci are under selection; hence, the low sporulation efficiency exhibited by S288C is acceptable (*Deutschbauer and Davis, 2005*). In contrast, the final haploid cells in the ubiquitin SGA must contain all modified ubiquitin loci, haploid selectable markers, and a gene deletion, meaning that an exceedingly small percentage of spores will have the desired genotype. A strategy to ameliorate this problem would be to increase the efficiency of sporulation, thereby both reducing the number of unsporulated diploid cells (background), and increasing the number of correct haploid spores. Overexpression of Ime2, a positive regulator of sporulation (*Neiman, 2011*), was able to increase the sporulation efficiency of S288C moderately, from 12% in wild-type diploids, to 38% (*Figure 1—figure supplement 5*). Other genetic manipulations similarly led to only modest increases in sporulation efficiency (*Figure 1—figure supplement 5*). The SK1 strain background provided a more efficient and physiologic solution, as it exhibits 92% sporulation efficiency (*Deutschbauer and Davis, 2005*; *Ben-Ari et al., 2006*). The high sporulation efficiency also allowed for the omission of one haploid selection step commonly used in SGA studies, *lyp1Δ* selection (*Figure 1—figure supplement 4*, Materials and methods). Hence, all K-to-R ubiquitin mutant strains and a novel single-gene deletion library were generated in the SK1 strain background (*Figure 1A*, Materials and methods). To further reduce the number of selections in the ubiquitin SGA protocol and increase the percentage of spores carrying all desired mutations following sporulation, a *ubi1* allele lacking the ubiquitin gene was crossed into both the ubiquitin K-to-R strains and the gene deletion array, thereby eliminating an additional selection step (*Figure 1A*, *Figure 1—figure supplement 6A–B*, Materials and methods).

The study of quantitative genetic interactions has proven to be a powerful approach to uncover novel functions of genes (*Schuldiner et al., 2005*; *Braberg et al., 2013*; *Costanzo et al., 2016*). Negative (synthetic) genetic interactions, which arise when a double mutant exhibits a more severe phenotype than is expected from the corresponding single mutant phenotypes, often occur when the gene pair functions in parallel or redundant pathways. Positive genetic interactions, defined as

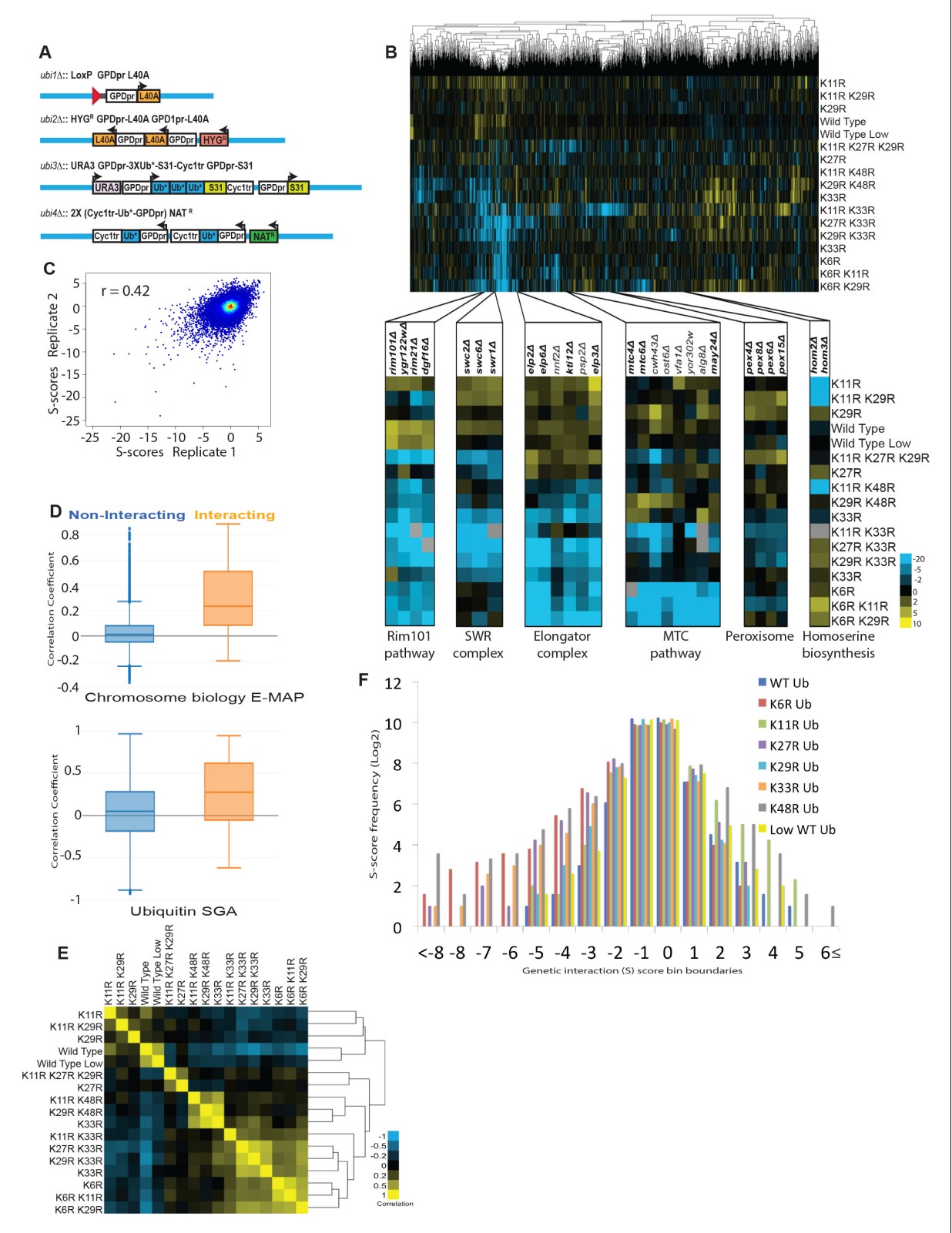

**Figure 1.** The ubiquitin linkage SGA reveals the genetic interactome of polyubiquitin chain types and functional relationships between lysines of ubiquitin. (**A**) *UBI1-UBI4* were modified to express lysine to arginine mutant ubiquitin alleles under the control of the constitutive GPD promoter. Ribosomal proteins, which are normally encoded at *UBI1-UBI3*, were also placed under the control of the GPD promoter individually, or as ubiquitin-fusions. (**B**) *Top:* A clustered heat map of quantitative genetic interactions between ubiquitin K-to-R mutants and an array of single gene

*Figure 1 continued on next page*

*Figure 1 continued*

deletions. Each K-to-R is expressed as the only ubiquitin present in the cell, except for K48R, which is lethal and was therefore co-expressed with WT ubiquitin (20% of total). As indicated in the adjacent color legend, blue pixels represent negative genetic interaction scores, yellow pixels represent positive genetic interaction scores, and black pixels indicate no interaction. Grey pixels represent interactions displaying extremely low signal, which could represent either strong negative interactions or pinning errors. *Bottom:* Unbiased hierarchical clustering identifies known gene modules. Named genes are part of the complexes or pathways listed (bold), or have related functions. (C) Scatter plot comparing the genetic interaction scores in two biological replicates of the ubiquitin SGA. The correlation coefficient between the two replicates was calculated. (D) The correlation of genetic interaction profiles of physically interacting or non-interacting gene pairs in the ubiquitin SGA and the chromosome biology E-MAP (*Collins et al., 2007*). (E) The correlation between the genetic interactomes of K-to-R ubiquitin mutants is shown as a heat map of correlation coefficients, and a dendogram showing hierarchical clustering. As the color legend indicates, correlation coefficient values of 1 are shown as yellow pixels, while values of −1 are shown as blue pixels. (F) A histogram showing the frequencies (Log2) of genetic interactions with the indicated S-scores for each single K-to-R ubiquitin mutant.

DOI: https://doi.org/10.7554/eLife.42955.002

The following figure supplements are available for figure 1:

**Figure supplement 1.** Engineering of *ubi2-4* to express K-to-R mutant ubiquitin.

DOI: https://doi.org/10.7554/eLife.42955.003

**Figure supplement 2.** Analysis of ubiquitin levels in K-to-R ubiquitin mutants.

DOI: https://doi.org/10.7554/eLife.42955.004

**Figure supplement 3.** Engineering of strains expressing low levels of ubiquitin, and K48R ubiquitin mutants.

DOI: https://doi.org/10.7554/eLife.42955.005

**Figure supplement 4.** The ubiquitin allele SGA protocol Ubiquitin K-to-R mutant strains were mated to an array of single-gene deletion strains.

DOI: https://doi.org/10.7554/eLife.42955.006

**Figure supplement 5.** Genetic modifications modestly increase the sporulation efficiency of the S288C yeast strain.

DOI: https://doi.org/10.7554/eLife.42955.007

**Figure supplement 6.** Deletion of *UBI1* in K-to-R ubiquitin mutant strains and the in the SK1 gene deletion array.

DOI: https://doi.org/10.7554/eLife.42955.008

situations in which a double mutant has a phenotype equal to (epistasis) or less severe than the sickest single mutant (suppression), can occur when both genes function in the same pathway.

The ubiquitin SGA identified thousands of candidate genetic interactions between gene deletions and K-to-R ubiquitin mutations (*Figure 1B*). While several other large SGA screens have analyzed genetic interactions between large sets of gene deletions with known functional relationships, the ubiquitin SGA was carried out using a small number of ubiquitin mutants and an unbiased gene deletion array. Because genes with known related functions are more likely to have genetic interactions, it was expected that the ubiquitin SGA would have a lower density of true genetic interactions relative to other studies. This has the effect of decreasing the reproducibility of any single genetic interaction score. Nonetheless, two biological replicates of the ubiquitin SGA had a correlation coefficient of 0.42, in line with other large genetic interaction screens (*Figure 1C*).

To characterize the robustness of the ubiquitin SGA further, the correlation between genetic and known physical interactions among genes was analyzed. Genes whose protein products are known to physically interact are generally also more likely to share genetic interactions and thus tend to have highly correlated genetic interaction profiles. The enrichment of genetic interaction similarities among gene pairs whose products physically interact was compared to protein pairs that have not been reported to interact. As expected, the median correlation coefficient within physically interacting gene pairs is significantly higher than among non-interacting gene pairs (*Figure 1D*). Although the genetic profiles of gene deletions in the ubiquitin SGA include only 17 queries (i.e. the lysine mutants), the screen is similarly enriched for genetic similarity within physically interacting gene pairs when compared to the published chromosome biology E-MAP, wherein the genetic interaction profiles contain 754 queries and is biased for genes with known related functions (*Figure 1D*). Together, these analyses demonstrate the robustness and biological significance of the ubiquitin SGA.

Unbiased hierarchical clustering orders the mutant strains in the ubiquitin SGA relative to each other based on the similarity of their genetic interactions (*Figure 1B*). Genes that function in the same pathways or have similar functions generally have common genetic interactions. As expected, the ubiquitin SGA identified known gene modules based on their genetic interactions with the K-to-R ubiquitin mutants. For example, *rim101Δ*, *rim21Δ*, *ygr122wΔ*, and *dfg16Δ* form a cluster in the ubiquitin SGA (*Figure 1B*). All four genes are members of the Rim101 pathway, whose

eponymous gene is a transcription factor that is activated in response to alkaline conditions in a manner dependent on Rim21, Ygr122Wp, and Dfg16 (*Maeda, 2012*). Other examples of known gene modules that were clustered by the ubiquitin linkage SGA include (*Figure 1B*): Three subunits of the chromatin-remodeling SWR-complex, *SWC2, SWC3*, and the catalytic subunit *SWR1* (*Rando and Winston, 2012*); several genes in the MTC-gene cluster (*MTC4, MTC6, MAY24*) that was recently implicated in trafficking through the secretory pathway (*Costanzo et al., 2016*), as well as genes with related functions in N-linked glycosylation (*OST6, ALG8*) (*Knauer and Lehle, 1999*; *Stagljar et al., 1994*), endocytic protein trafficking (*VFA1*) (*Arlt et al., 2011*), or lipid biosynthesis (*YOR302W*) (*Delbecq et al., 2000*); several members of the transcription and translation-regulating Elongator complex (*ELP2, 3, 6*) and *KTI12*, a known binding partner (*Krogan and Greenblatt, 2001*; *Petrakis et al., 2005*), as well as two genes, *NNF2* and *PSP2*, with putative related roles in transcription and splicing, respectively (*Briand et al., 2001*; *Waldherr et al., 1993*); four genes, *PEX4, 8, 6, 15* that function in peroxisome biogenesis and trafficking of proteins between peroxisomes and the cytosol (*Sibirny, 2016*); and finally, two genes, *HOM2* and *HOM3*, that participate in the homoserine biosynthesis pathway (*Mountain et al., 1991*).

The K-to-R ubiquitin mutants were also clustered based on the similarity of their genetic interactions with the gene deletion array. As expected, clustering of strains with common K-to-R mutations is generally observed (*Figure 1E*). For example, strains carrying the K33R mutation are clustered and have high correlation coefficients among each other. The same can be said of strains carrying the K6R and K48R mutations (*Figure 1E*). Interestingly, the K6R and K33R clusters are most similar to each other, suggesting similar physiological effects of mutating those ubiquitin lysines (*Figure 1E*). A surprising feature of the ubiquitin allele SGA is that the K11R mutation, despite the high abundance of K11 linkages in cells (*Xu et al., 2009*), does not drive clustering of double and triple mutants carrying K11 mutations (*Figure 1E*). Clustering is driven by a combination of the number and strength of common interactions between K-to-R mutants. Indeed, the K11R ubiquitin mutant has relatively few strong genetic interactions, suggesting that K11 linkages have a narrow set of functions in yeast (*Figure 1F*).

The ubiquitin SGA uncovered thousands of candidate genetic interactions between gene deletions and ubiquitin lysine mutations. To examine a few interactions in more detail, gene deletion strains were mated to all single lysine-to-arginine mutants, as well as wild-type ubiquitin strains. Following tetrad dissection, the growth phenotypes of single and double mutant cells were assessed. K63R mutants are included in this targeted analysis. Genes exhibiting genetic interactions with a single linkage type were examined first. The negative genetic interactions observed between *ubc4Δ* and K27R and between *hom2Δ* and K11R proved to be reproducible and specific (*Figure 2A and B*). Next, *CDC26*, a gene whose deletion exhibited genetic interactions with multiple linkages (K48R >> K11≥K33R), was analyzed. Deletion of *CDC26* in combination with mutation of ubiquitin's K11 or K48 caused significant synthetic growth defects, whereas only a mild negative interaction was observed in *cdc26Δ* and K33R double mutant cells (*Figure 2C*). Having demonstrated the robustness and reproducibility of the ubiquitin SGA, the physiological functions of ubiquitin linkage types were explored.

## K11 ubiquitin linkages promote amino acid homeostasis

The K11R mutant was of special interest due to the fact that K11-linkages are the most abundant non-K48 linkages in *S. cerevisiae* (*Xu et al., 2009*), and because of their important function in metazoan cell cycle regulation (*Meyer and Rape, 2014*; *Matsumoto et al., 2010*; *Wu et al., 2010*). Characterization of the K11R mutant genetic interactome using gene ontology (GO) term enrichment analysis revealed a striking enrichment of genes involved in amino acid biosynthesis (*Figure 3A*) (*Huang et al., 2009a*; *Huang et al., 2009b*), suggesting a possible role of K11 linkages in amino acid homeostasis. To identify genes that had interactions specifically with K11R and not with other K-to-R ubiquitin mutations, the K11R mutant genetic interactome was compared to the averaged interactome of all single and double mutant strains not containing the K11R mutation (*Figure 3B*). Deletions of *HOM2* or *HOM3*, two genes involved in the biosynthesis of homoserine, a threonine and methionine precursor (*Mountain et al., 1991*), and to a lesser extent deletion of *GLY1*, which encodes an enzyme that generates threonine and glycine (*Liu et al., 1997*), exhibited strong genetic interactions specifically with the K11R ubiquitin mutants (*Figure 3B and C*). These genetic interactions suggested the existence of functional redundancy between K11 linkages and threonine or

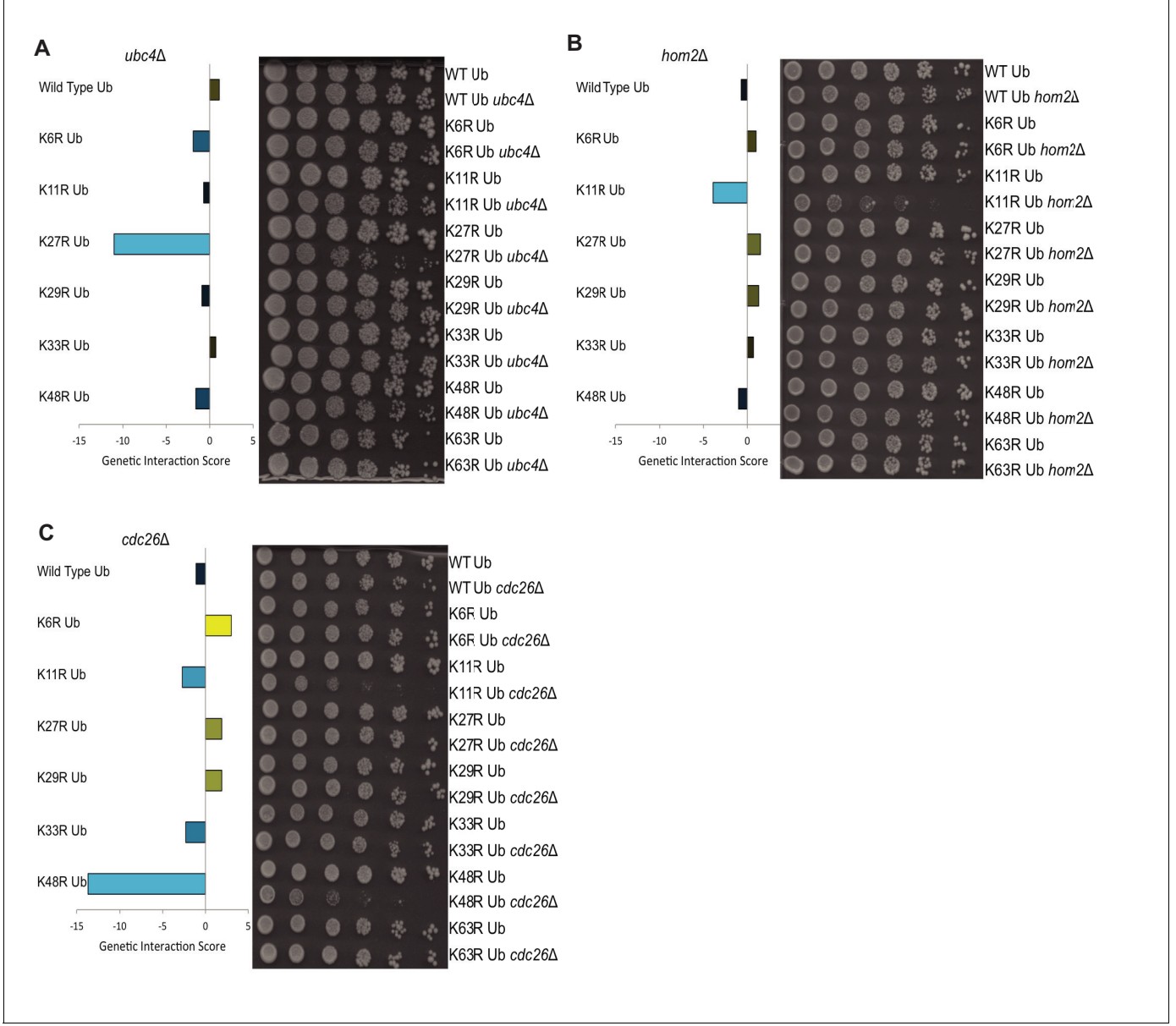

**Figure 2.** Gene deletions exhibit genetic interactions with specific ubiquitin linkage types. (A–C) *Left:* The genetic interaction scores of K-to-R ubiquitin mutants with deletions of *UBC4* (A), *HOM2* (B), or *CDC26* (C); Scores less than zero represent varying degrees of negative genetic interactions (blue bars), while scores larger than zero indicate positive interactions (yellow bars). *Right:* Candidate genetic interactions were validated by 5-fold spot dilution assays of the indicated strains.

DOI: https://doi.org/10.7554/eLife.42955.009

methionine biosynthesis. To determine whether perturbations of threonine or methionine levels was the basis for the synthetic growth defect observed in K11R and *hom2Δ* or *hom3Δ* double mutant cells, the growth medium was supplemented with homoserine, excess threonine, or methionine. Rescue of the synthetic growth defect by either homoserine or threonine suggested that K11-linked polyubiquitin chains function redundantly with threonine biosynthesis (*Figure 3D*).

Whereas negative genetic interactions point to redundancy between genes and ubiquitin linkage types, the deletions of genes that function in the same pathway as a specific ubiquitin linkage are likely to have genetic interactions that are similar to that linkage type. Comparison of a comprehensive yeast genetic interaction network to the genetic interactome of the K11R ubiquitin mutant revealed that it is most similar to deletions of *GNP1* (*Ryan et al., 2012*), a high-affinity threonine

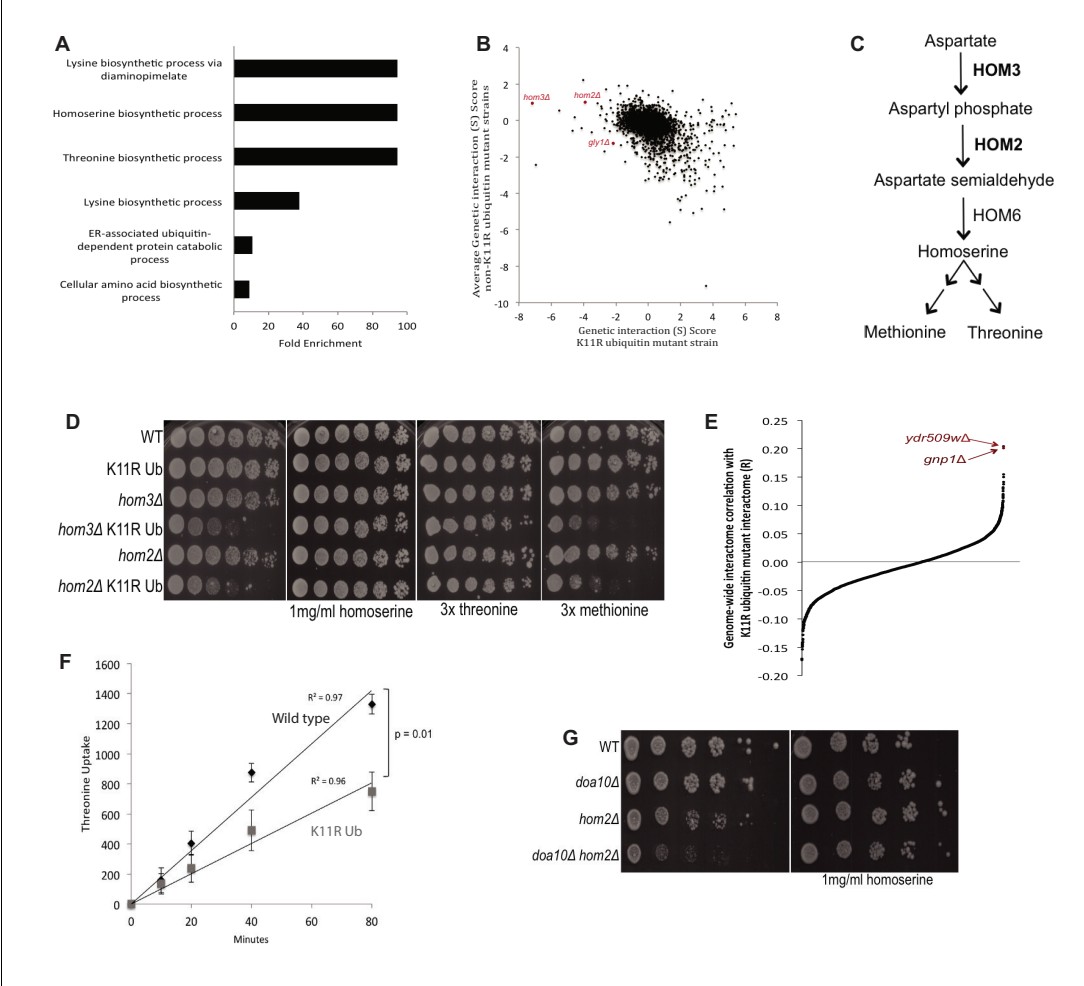

**Figure 3.** K11 linkages are functionally redundant with threonine biosynthesis and promote threonine import. (**A**) Gene Ontology (GO) term enrichment analysis was performed on the genetic interactions of the K11R ubiquitin mutant having genetic interaction scores of −2 or less. The fold enrichment of biological function GO terms (BP_Direct) that were significantly enriched is shown. (**B**) The genetic interaction scores of the K11R ubiquitin mutant were plotted against the averaged genetic interaction scores of all K-to-R strains not containing the K11R mutation. Highlighted genes had negative interactions specifically with the K11R ubiquitin mutant, and had related functions in amino acid biosynthesis. (**C**) *HOM2* and *HOM3* are required for the biosynthesis of homoserine, a methionine and threonine precursor. (**D**) The negative genetic interactions of the K11R ubiquitin mutant with deletions of *HOM2* or *HOM3* were validated by fivefold spot dilution assays of the indicated strains. Excess homoserine, threonine, or methionine was added to C-leu plates as noted. (**E**) The correlation coefficient between the genetic interactions of the K11R ubiquitin mutant and single gene deletion strains was calculated. The genetic interactome of the K11R ubiquitin mutant is most similar to the genetic interactions of strains carrying deletions of *GNP1* or *YDR509W*. (**F**) The import of tritiated threonine into wild-type and K11R ubiquitin mutant cells was measured by scintillation counting. The Y-axis shows the measured counts per minute (CPM) divided by optical density (OD600) of the cell cultures (CPM/OD600). (**G**) A negative genetic interaction between deletions of *HOM2* and *DOA10* is shown by spot dilution assays of the indicated strains. The genetic interaction is abrogated by addition of homoserine to C-leu plates.

DOI: https://doi.org/10.7554/eLife.42955.010

The following figure supplement is available for figure 3:

**Figure supplement 1.** Gnp1 is a major threonine permease.

DOI: https://doi.org/10.7554/eLife.42955.011

permease (*Figure 3—figure supplement 1*) (*Regenberg et al., 1999*), and *YDR509W*, a dubious open-reading frame that overlaps the N-terminus of *GNP1* and is thus likely to also abrogate *GNP1* expression (*Fisk et al., 2006*) (*Figure 3E*). The similarity between the K11R mutant and *gnp1Δ* genetic interactomes suggested that K11 linkages play a role in promoting threonine import. To assess that model, the import of radioactively labeled threonine into wild-type and K11R ubiquitin cells was measured, revealing a significant defect in the K11R ubiquitin mutant, and thus showing

that K11 linkages promote threonine import (*Figure 3F*). This previously unreported function of K11-linked polyubiquitin chains in amino acid import is likely the underlying basis of the synthetic growth defect observed in *K11R hom2Δ* or *K11R hom3Δ* cells.

Polyubiquitin chains are synthesized by E3 ubiquitin ligases (*Hershko and Ciechanover, 1998*). If a particular E3 ligase were involved in the K11-mediated import of threonine, it would be expected to also have negative genetic interactions with *hom2Δ* or *hom3Δ*. Deletion of *DOA10,* an endoplasmic-reticulum-resident E3 ubiquitin ligase that is responsible for approximately 50% of K11 linkages in yeast cells (*Zattas and Hochstrasser, 2015*; *Xu et al., 2009*), in combination with *hom2Δ* also led to a synthetic growth defect that was rescued by homoserine (*Figure 3G*), suggesting that Doa10, along with K11 ubiquitin chains, contribute to threonine import. These results suggest that the Doa10-mediated ubiquitination of a substrate, or a set of substrates, might regulate amino acid import. The Doa10-dependent ubiquitinome was analyzed using quantitative mass spectrometry and ubiquitin-remnant enrichment. While known Doa10 substrates were less ubiquitinated in cells lacking Doa10, or its cognate E2s Ubc6 and Ubc7, the substrate responsible for the role of Doa10 in amino acid homeostasis remains unknown (*Supplementary file 5*). Nonetheless, the discovery that Doa10 is an E3 ligase that contributes to the function of K11-linkages in amino acid import counters models suggesting the observed amino acid import defect is due to non-specific perturbations of ubiquitin.

## K11 ubiquitin linkages contribute to mitotic progression

The anaphase promoting complex (APC) is a highly conserved and essential E3 ubiquitin ligase that governs the metaphase to anaphase transition and mitotic exit by ubiquitinating several substrates, including securin and S and M-phase cyclins (*Sumara et al., 2008*; *Tyers and Jorgensen, 2000*; *Thornton et al., 2003*). In yeast, the APC cooperates with two E2 ubiquitin conjugases, Ubc4 and Ubc1, to decorate its substrates with polyubiquitin chains and target them for proteasomal degradation (*Rodrigo-Brenni and Morgan, 2007*). Ubc4 is a promiscuous E2 that monoubiquitinates or initiates short ubiquitin chains on substrates (*Arnason and Ellison, 1994*; *Kirkpatrick et al., 2006*). Ubc1 subsequently extends those initial modifications with long and homogenous K48-linked ubiquitin chains (*Rodrigo-Brenni and Morgan, 2007*; *Rodrigo-Brenni et al., 2010*). K48 of ubiquitin is necessary for processive APC-substrate ubiquitination in vitro (*Rodrigo-Brenni and Morgan, 2007*), and is the only linkage type required for normal cell cycle progression in vivo (*Rodrigo-Brenni et al., 2010*). At high temperatures, a non-essential subunit of the APC, Cdc26, aids in the assembly and promotes the stability, of the APC holoenzyme (*Schwickart et al., 2004*; *Zachariae et al., 1998a*). As expected given the well-established role of K48 linkages in APC-mediated proteasomal degradation, K48R ubiquitin mutant cells lacking Cdc26 exhibit a significant growth defect (*Figure 2C*). Interestingly, deletion of *CDC26* in K11R mutant cells also leads to a strong synthetic growth defect (*Figure 2C*), suggesting a possible contribution of K11 linkages to APC function.

Although K11-linked polyubiquitin chains had not been previously reported to function in APC-substrate degradation in yeast, their role in metazoan APC-substrate turnover is well understood (*Meyer and Rape, 2014*; *Williamson et al., 2009*; *Matsumoto et al., 2010*; *Wu et al., 2010*; *Jin et al., 2008*; *Yau et al., 2017*). Analogous to the yeast APC, the metazoan APC interacts with two E2 enzymes, Ube2C and Ube2S, that sequentially ubiquitinate substrates (*Wu et al., 2010*; *Brown et al., 2016*). Ube2c initially modifies substrates with monoubiquitin or short chains that are then elongated by Ube2S with K11-linked chains (*Wu et al., 2010*). The negative genetic interaction between K11R and *cdc26Δ* suggested that K11 linkages might also be important for the regulation of APC substrates in yeast. Indeed, cell cycle analysis experiments with cells lacking Cdc26 revealed that the cell-cycle-regulated levels of securin and several cyclins were altered in K11R ubiquitin mutants relative to cells expressing wild-type ubiquitin (*Figure 4A*). For example, the mitotic cyclin Clb2 persisted much longer during mitosis in K11R cells, with its disappearance being delayed by almost 30 min (*Figure 4A*). The levels of the S/M cyclin, Clb3, and of the Separase-inhibitor, Securin (Pds1), were also misregulated in K11R mutant cells, although to a lesser extent than Clb2 (*Figure 4A*). Clb5, an S-phase cyclin, is normally targeted for degradation during the M and G1 phases by the APC (*Ostapenko et al., 2015*; *Shirayama et al., 1999*). Its expression during both M and G1 was elevated in K11R mutant cells, suggesting that K11 was required for complete turnover of this substrate (*Figure 4A*).

To ascertain whether the misregulation of APC substrates was caused by a defect in their turnover, and not due to other effects of mutating K11, *cdc26Δ* cells were synchronized in G1 phase,

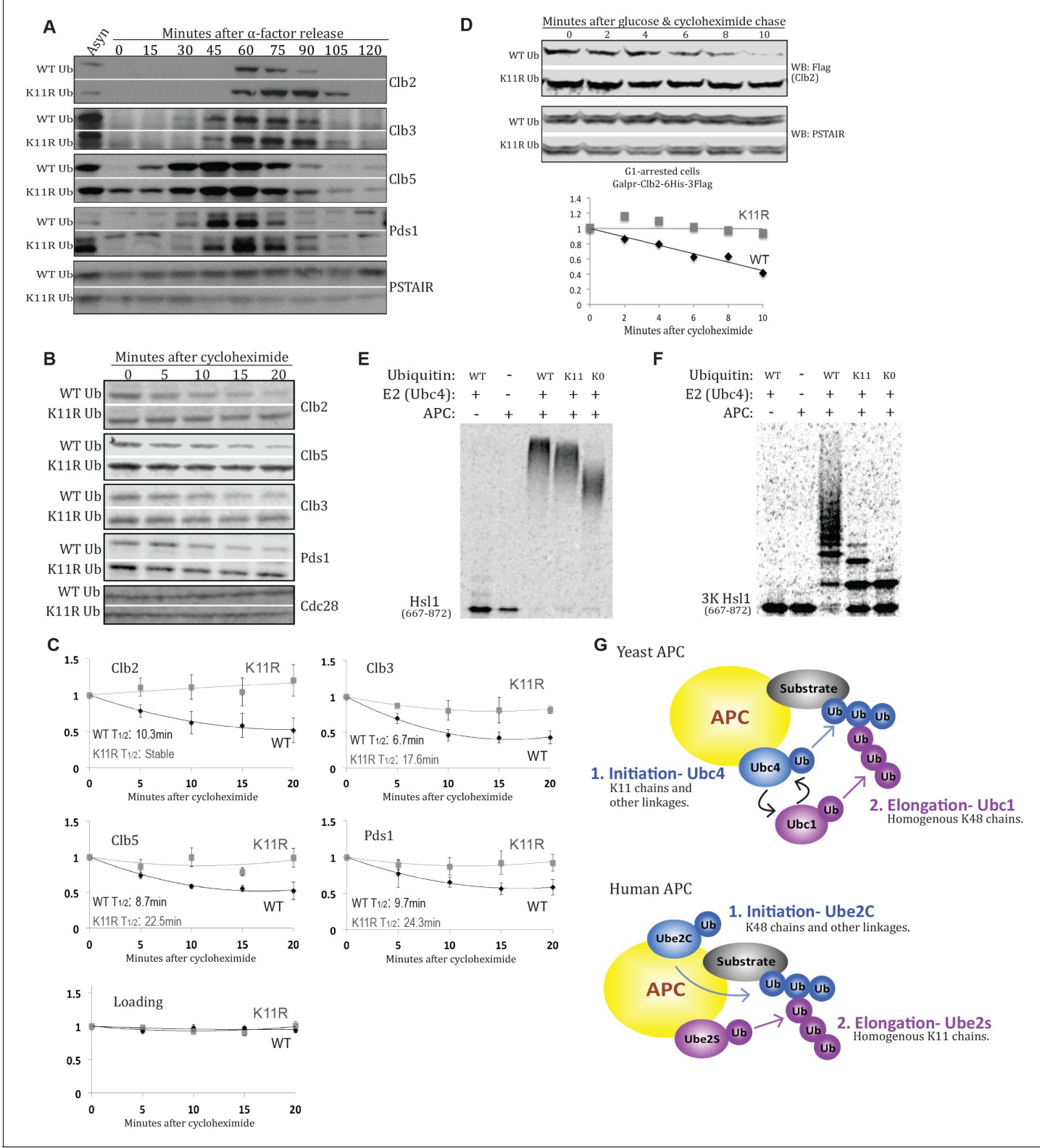

**Figure 4.** K11 linkages contribute to APC-substrate turnover. (**A**) Wild type and K11R ubiquitin mutant cells lacking Cdc26 were synchronized in G1 phase with α factor, then released into the cell cycle. Samples were taken at the specified times after release and analyzed by probing western blots for the indicated proteins. (**B, C**) *cdc26Δ* cells expressing wild type or K11R ubiquitin were arrested in G1 with α-factor. Cells were washed in fresh media to release from the arrest and, after sixty minutes, protein synthesis was blocked by the addition of cycloheximide. The APC was active at this time point,

*Figure 4 continued on next page*

*Figure 4 continued*

as measured by Pds1 and Clb5 turnover (Panel 4A). Samples were taken at the specified times after cycloheximide addition and analyzed by probing western blots for the indicated proteins. Bands were quantified using the Licor Image Studio Software and were normalized to Cdc28 or PSTAIR as loading controls (n = 3). A second order polynomial best-fit line was calculated. Half lives were calculated by determining the slope of the line tangent to the best fit curve at T = 0. (D) Cells were synchronized in G1 phase with α factor. During the G1 arrest, Clb2-6xHis-3xFlag expression was induced by the addition of galactose, and cells were shifted to 37°C. Clb2 synthesis was then blocked by removal of galactose, and addition of glucose and cycloheximide. The levels of Clb2-6xHis-3xFlag in wild-type and K11R ubiquitin cells lacking Cdc26 were analyzed by western blotting at the indicated times. Cdc28 was examined as a loading control using Anti-PSTAIR antibody. Bands were quantified as in *Figure 4C*. (E) A radioactively labeled fragment of Hsl1, amino acids 667–872, was incubated with or without APC/Cdh1, Ubc4, and different forms of ubiquitin, as indicated. The ubiquitination of Hsl1 was analyzed by SDS-PAGE and autoradiography. (F) All lysine residues on Hsl1 (667-872) were mutated to arginine, except for two lysines (K775, 812) that are part of 2 KEN box degrons, and lysine 747, which has been previously reported as being ubiquitinated. This Hsl1 fragment is termed Hsl1-3K, and was analyzed as in 4E. (G) A model describing the well-established role of K11 linkages in mammalian cells, and their function in yeast APC substrate degradation as suggested by the data presented here.

DOI: https://doi.org/10.7554/eLife.42955.012

The following figure supplement is available for figure 4:

**Figure supplement 1.** Cell cycle profiling by FACS.

DOI: https://doi.org/10.7554/eLife.42955.013

and the half life of APC substrates was examined 60 min after release from G1, when the APC was active as evidenced by securin turnover (*Figure 4A*; *Figure 4—figure supplement 1A*). The turnover of all examined substrates is delayed in K11R mutant cells (*Figure 4B,C*). Interestingly, the turnover of substrates stalls after approximately 10 min in cycloheximide, likely due to degradation of APC substrate adaptors (*Foe et al., 2011*). The half-lives of substrates were estimated by fitting a second-order polynomial trendline to the data and subsequently calculating the slope of the tangent line at t = 0 min. On average, the half-lives of examined substrates were extended by approximately 2.5 fold in the K11R ubiquitin mutant (*Figure 4C*). Because defects in cell cycle progression in K11R mutant cells could explain the delay in APC substrate turnover, it was necessary to determine the half-lives of APC substrates independently of cell cycle progression. Ectopic expression of Clb2 in G1-arrested cells, which have active APC (*Zachariae et al., 1998b*), permitted the analysis of the Clb2 half-life in cells that were in the same cell cycle stage. The half-life of Clb2 was analyzed after addition of cycloheximide in G1 phase, revealing a delay in Clb2 turnover in K11R mutant cells (*Figure 4D*, *Figure 4—figure supplement 1B*).

To examine K11 chain usage by the APC biochemically, the capacity of the yeast APC to modify substrates with K11-linked polyubiquitin chains was analyzed in vitro. The ubiquitination of an in vitro-translated fragment of the APC substrate Hsl1, by APC-Cdh1 and Ubc4, the initiating APC-cognate E2, was assayed in the presence of either WT, K0 (K6,11,27,29,33,48,63R), or K11-only (K6,27,29,33,48,63R) ubiquitin. Under these conditions, Hsl1 is strongly ubiquitinated in the presence of WT ubiquitin and, to a slightly lesser extent, in reactions containing K11-only ubiquitin (*Figure 4E*). In contrast, Hsl1 ubiquitination is significantly reduced in reactions containing K0 ubiquitin, as compared to K11-only. The ubiquitin smear observed when Hsl1 is incubated with K0 ubiquitin likely represents the attachment of monoubiquitins to several of the 21 lysines in the Hsl1 fragment used in the experiments. Significantly, the slower electrophoretic mobility of Hsl1 modified with K11-only ubiquitin relative to K0 ubiquitin shows that the yeast APC extends monoubiquitin modifications of Hsl1 with K11-linked ubiquitin chains in vitro.

To ascertain the length of homogenous K11-linked chains generated by the yeast APC, all lysines of Hsl1 were mutated to arginine, except for two lysines that constitute two KEN-box degrons and K747 of Hsl1, which has been previously reported to be ubiquitinated. This mutant form of Hsl1, termed Hsl1-3K, allowed for the detection of Hsl1 ubiquitination at a single lysine residue (K747). The observation that Hsl1-3K is decorated with a single monoubiquitin in reactions containing K0 ubiquitin confirmed that a one lysine, likely K747, is ubiquitinated by the APC in Hsl1-3K (*Figure 4F*). The reduced level of ubiquitination observed for Hsl1-3K relative to WT Hsl1 (compare the loss of unmodified substrate in lane 5 of *Figure 4E and F*) suggests that recognition of this mutant substrate by the APC is strongly compromised, which is likely to compromise the enzyme's processivity significantly. Nonetheless, the substrate is polyubiquitinated in the presence WT ubiquitin (*Figure 4F*). Significantly, the monoubiquitination of Hsl1-3K in the presence of K11-only ubiquitin is

extended with at least two further ubiquitins, leading to a chain composed of at least three ubiquitin protomers that are necessarily linked through lysine-11 of ubiquitin (*Figure 4F*). The higher level of Hsl1-3K ubiquitination seen with WT relative to K11-only ubiquitin (*Figure 4F*, lanes 3 and 4) could either represent ubiquitin chain branches or the combinatorial effect of a highly mutated ubiquitin and substrate. Altogether, our results show that K11 linkages contribute to APC-substrate turnover in yeast.

## Discussion

Although some large-scale proteomics studies have advanced our understanding of the physiological roles of specific polyubiquitin chain types, much of the progress has occurred through the study of single pathways. A significant challenge for the ubiquitin field has been the partial redundancy between ubiquitin linkage types (*Xu et al., 2009*). The analysis of genetic interactions, however, reveals phenotypic functional relationships between genes, thereby facilitating the discovery of physiologically important functions of genes. The synthetic genetic array (SGA) reported here represents the first comprehensive genetic analysis of lysine-to-arginine ubiquitin mutant alleles. This SGA leverages the power of yeast genetics to uncover physiologically important functional relationships between ubiquitin linkage types and thousands of genes.

The ubiquitin linkage SGA necessitated the development of a novel SGA protocol and reagents to permit the analysis of genetic interactions in cells carrying five mutations, as well as haploid selectable markers. The large number of loci under selection meant that an extremely small number of spores carried the desired genotype for SGA analysis. To increase the number of spores, the ubiquitin linkage SGA was carried out in the SK1 strain background, which exhibits about eightfold higher sporulation efficiency relative to S288C (*Deutschbauer and Davis, 2005*), the yeast strain most commonly used in SGA studies. This also permitted the omission of a haploid selection step by decreasing background unsporulated diploids. The ubiquitin linkage SGA therefore required the development of a new gene deletion library in the SK1 background. To increase the percentage of spores carrying the desired genotype for SGA analysis, the *UBI1* locus was replaced in the entire SK1 deletion library, as well as in the panel of K-to-R ubiquitin mutant strains, thereby reducing the loci under selection. Ultimately, the technical advancements described here reduced, from eight to six, the number of loci under selection, thereby making the ubiquitin linkage SGA feasible.

A novel role of K11 linkages in promoting amino acid import adds to already well-established roles of the ubiquitin-proteasome system in that process. First, expression of amino acid permeases is promoted by the transcription factors Stp1/2, which are proteolytically processed into their active forms in a ubiquitin-dependent process utilizing K48 ubiquitin linkages (*Ljungdahl, 2009*; *Petroski and Deshaies, 2005*). We found that neither this pathway nor the levels or localization of the threonine permease Gnp1 are affected by the K11R ubiquitin mutation (data not shown). Conversely, downregulation of permeases occurs when an essential E3 ligase, Rsp5, modifies their intracellular domains with monoubiquitin and short K63-linked polyubiquitin chains, thereby triggering endocytosis and vacuolar targeting (*MacGurn et al., 2012*; *Lauwers et al., 2010*; *Kim and Huibregtse, 2009*). The observed enrichment of threonine biosynthesis genes in the genetic interactions of the K11R ubiquitin mutant would suggest a specific function in threonine import. A more likely model, however, is that threonine homeostasis is more susceptible to perturbations, perhaps due to relatively low redundancy in its modes of import (*Regenberg et al., 1999*). Thus, our data may point to a role for K11 linkages with many importers whose functional impairment is not phenotypic.

The ubiquitin allele SGA also revealed a previously unreported contribution of K11 linkages to cell cycle regulation. The genetic interaction between K48 linkages and *CDC26* was not unexpected, as it is understood that the yeast APC decorates its substrates with K48-linked ubiquitin chains (*Rodrigo-Brenni et al., 2010*). The genetic interaction between K11 linkages and *CDC26* identified by the ubiquitin allele SGA led to the recognition that K11 linkages also contribute to APC-substrate turnover in vivo, and that the APC can directly modify a substrate with K11-linked polyubiquitin in vitro. These results suggest that the metazoan APC, which modifies its substrates with K11-linked ubiquitin chains (*Meyer and Rape, 2014*; *Williamson et al., 2009*; *Matsumoto et al., 2010*; *Wu et al., 2010*; *Jin et al., 2008*; *Yau et al., 2017*), and the yeast APC are more alike than previously appreciated.

The metazoan and yeast APC both interact with two E2 enzymes that sequentially modify substrates (*Meyer and Rape, 2014*; *Wu et al., 2010*; *Rodrigo-Brenni and Morgan, 2007*; *Garnett et al., 2009*). The initiating E2 enzymes, Ube2C in metazoans and Ubc4 in yeast, modify substrates with single ubiquitins and short chains (*Kirkpatrick et al., 2006*). Because initiating E2s must be able to modify lysines on substrates, which can exist in diverse structural and chemical environments, they necessarily lack strong lysine preference (*Arnason and Ellison, 1994*; *Kirkpatrick et al., 2006*). Ubc4 or Ube2c-mediated modifications are then elongated by Ube2S in metazoans or Ubc1 in yeast. In contrast to initiating E2s, Ube2S and Ubc1 exhibit a strong preference for ubiquitin's lysine 11 and 48, respectively (*Rodrigo-Brenni et al., 2010*; *Wickliffe et al., 2011*), and are unable to efficiently modify lysines on substrates (*Wu et al., 2010*; *Rodrigo-Brenni and Morgan, 2007*). In light of preceding APC studies, the genetic, molecular, and biochemical results presented here suggest that the yeast APC modifies its substrates with a short, Ubc4-mediated heterogeneous base, in which K11-linked chains are most functionally important (*Kirkpatrick et al., 2006*). Ubc1 then extends those initial modifications with long K48-linked polyubiquitin chains (*Rodrigo-Brenni and Morgan, 2007*; *Rodrigo-Brenni et al., 2010*). This model (*Figure 4G*) differs from the well-established mechanism of the metazoan APC, which modifies substrates with a base of short chains with K48 and other linkages, which then branch into long K11-linked chains. The fact that the contribution of K11 linkages to APC-substrate turnover is appreciable only in cells without Cdc26 suggests a nuanced role of K11 linkages in yeast, similar to metazoans, wherein disruption of the gene encoding the K11-specific E2, Ube2s, is not lethal. Pivotal work has shown that K11/K48 branched chains significantly enhance substrate degradation in mammalian cells (*Meyer and Rape, 2014*; *Yau et al., 2017*). Whether this holds true in yeast has not been studied, but would provide an appealing explanation for the use of K11-linkages in APC function. An appealing model is that this allows turnover of substrates that must be eliminated immediately and synchronously, like securin, to have priority over less urgent quality control activities.

Detailed analysis of genetic interactions uncovered by the ubiquitin linkage SGA analysis has enabled us to discover two novel physiological functions of K11 ubiquitin linkages in two important processes: amino acid transport and cell cycle regulation. Altogether, the ubiquitin allele SGA has broadened our global understanding of the functions of polyubiquitin chain types by revealing their genetic interactomes, and demonstrated the utility of genetic analysis for interrogating the complexity of the ubiquitin-proteasome system.

## Materials and methods

### Generation of an SK1 gene deletion library

SK1 diploids of genotype *his3/his3 ura3/ura3* CAN1/*can1::STE2pr-SpHIS5* were transformed with a PCR-generated cassette containing the KanMX4 module flanked by ORF-specific homology. Transformants were selected by their dominant resistance to geneticin (G148). Correct replacement of the target locus with the KanMX4 module was verified by the appearance of PCR products of expected size. The resulting heterozygotes were sporulated at 25°C for 2 days. To select for haploids of MATa, and carrying the *STE2pr-SpHIS5* at the *can1* locus, cells were pinned onto SD +canavanine + uracil – histidine plates and incubated at 30°C for 2 days. To select for cells carrying single-gene deletions, haploids were then transferred to plates containing 400 mg/L G418 plates and incubated at 30°C for 24 hr.

### Deletion of *UBI1*

The ubiquitin linkage SGA required the analysis of genetic interactions in cells carrying mutations in all four ubiquitin loci, a single-gene deletion, as well as haploid selectable markers. To decrease the number of loci under selection, and thereby increase the percentage of spores with the desired genotype, the *UBI1* locus was deleted in all strains in the SK1 Deletion library, as well as in the panel of K-to-R ubiquitin mutant strains. *UBI1* was replaced with a construct expressing the ribosomal protein Rpl40A from the constitutive GPD promoter (*Figure 1—figure supplement 6*). This decreased the number of selection steps required for the ubiquitin linkage SGA to 6: three ubiquitin loci, one gene deletion, two haploid and mating type selection markers.

As outlined in *Figure 1—figure supplement 6*, a 500 μL liquid culture of strain DS1 of the geno-type Mat alpha; *his3; ura3; CAN1; ubi1Δ::LoxP-GAL1pr-Cre-URA3-LoxP-GPDpr-RPL40A* was spread on a C-uracil plate to generate a lawn of cells that was subsequently pinned into the 384 format to allow mating to the SK1 gene deletion library. It is worth noting that the *GAL1pr-Cre* construct could not be maintained in bacteria due to leakiness of the promoter, and was therefore amplified from yeast genomic DNA directly. After mating to the SK1 gene deletion array on YPAD medium, diploids were selected using C-uracil +G418 plates. After sporulation for 2 days, haploids were selected first on C-uracil for 24 hr; then on C-uracil +G418 for 2 days. Haploids were then plated on 2% galactose containing plates for 2 days to induce *Cre* recombinase that deleted the endogenous ubiquitin in *UBI1,* as well as *URA3* and the *Cre* recombinase due to the position of the LoxP sites. The modified library was plated on 5-fluoro-orotic acid (FOA) plates and incubated at 30°C for 24 hr. This serves as a negative selection for the *URA3* auxotrophic marker, such that cells containing a functional *URA3* gene will die on FOA plates (*Boeke et al., 1984*). FOA negative selection was performed con-secutively three times to ensure *URA3* deletion. The modified *ubi1* locus was amplified from the library using primers outside the homology arms and sequenced to confirm the successful recombi-nation event. Deletion of *UBI1* did not result in changes in the total ubiquitin levels or doubling time, in agreement with a previous report (*Hanna et al., 2003*).

## Generation of K-to-R ubiquitin mutant strains

To express K-to-R mutant ubiquitin alleles from a strong, constitutive promoter, *TEF1, HYP2, PYK1, PDC1,* and GPD promoters were analyzed, and the GPD promoter was chosen since it resulted in the highest ubiquitin expression in agreement with a previous report (*Partow et al., 2010*).

Plasmids generated, and the relevant cloning information, are listed in *Supplementary file 1*. Generation of the *ubi1* allele is discussed above.

The *UBI2* locus was replaced with two tandem copies of the GPD promoter driving expression of the ribosomal protein Rpl40A (*Figure 1—figure supplement 1*). The modified locus was marked with the Hygromycin resistance gene. Deletion of *UBI2* resulted in a doubling time of 2 hr, compared to 1.6 hr for wild-type cells, in agreement with a previous report (*Jackson and Durocher, 2013*). The modified *ubi2* locus with two tandem GPDpr-*RPL40A* copies exhibited a normal doubling time.

The endogenous *UBI3* locus encodes a hybrid of ubiquitin fused to the ribosomal protein gene *RPS31*. The locus was replaced with a construct expressing a polypeptide composed of three tan-dem ubiquitin copies fused to one *RPS31* copy, under the control of the GPD promoter. An addi-tional *RPS31* gene was placed under the control of the GPD promoter at this locus (*Figure 1—figure supplement 1*). Several *UBI3* modifications were attempted prior to selecting this construct. Deletion of *UBI3* resulted in a doubling time of 7 hr, in agreement with a previous report (*Jackson and Durocher, 2013*). Multiple *ubi3* alleles were generated with varying numbers of GPDpr-*RPS31* copies. Although there was a correlation between the number of *RPS31* copies and doubling time, even five copies of GPDpr-*RPS31* resulted in a slow doubling time of 3 hr. It is sug-gested the ubiquitin fusion protein assists in ribosomal protein expression (*Finley et al., 1989*). Therefore, given the possibility that ubiquitin acts as a chaperone for Rps31, three tandem copies of ubiquitin were fused to *RPS31*. An additional GPDpr-*RPS*31 copy was integrated, and the *ubi3* locus was marked with *URA3* (*Figure 1—figure supplement 1*). The resulting *ubi3* allele led to growth rates similar to wild type.

The *UBI4* locus normally encodes five tandem copies of ubiquitin, expressed as single polypep-tide. *UBI4* was replaced with a two copies of a construct expressing, under the control of the GPD promoter, ubiquitin including the C-terminal asparagine found in endogenous *UBI4* and which is pro-teolytically processed to release mature ubiquitin (*Finley et al., 2012*; *Ozkaynak et al., 1987*). This locus was marked with a Nourseothricin (NAT) resistance cassette (*Figure 1—figure supplement 1*).

A strain expressing low levels of wild-type ubiquitin was included to control for the potential effects of altered ubiquitin levels. The strain differs from other strains in this study at *ubi4*, which is replaced only with the NAT resistance cassette. Thus, ubiquitin is only expressed from *ubi3* (*Fig-ure 1—figure supplement 3A*).

Lysine 48 of ubiquitin is essential, therefore, the K48R ubiquitin mutant strain was modified to express one copy of wild-type ubiquitin from the GPD promoter-driven triple-ubiquitin fusion at *ubi3* (*Figure 1—figure supplement 3B*). Double mutants carrying the K48R mutation and another lysine mutation expressed one copy of double lysine mutant ubiquitin (K11R K48R, for example) and two

copies of single mutant ubiquitin (K11R, for example) at *ubi3*. Double mutant ubiquitin was expressed from *ubi4*.

To integrate the constructs described above, a diploid strain homozygous for the engineered *ubi1* locus was transformed with the *ubi2*-targeting construct. Transformants were selected on hygromycin plates, and then transformed with the *ubi3*-targeting constructs carrying the various ubiquitin mutations. The resulting transformants were selected on plates lacking uracil, then sporulated. Triple mutant *ubi1; ubi2; ubi3* cells of mating type alpha were selected by growth on plates lacking uracil, and containing hygromycin. The *ubi4* targeting constructs carrying ubiquitin mutations were introduced into a wild-type diploid strain. The strain was sporulated, and *ubi4* mutant haploids of mating type a were selected by their growth on nourseothricin-containing plates. *ubi1; ubi2; ubi3* cells were mated to cells carrying the *ubi4* mutant locus. Diploids were sporulated, and haploids carrying all modified ubiquitin loci were selected.

## Optimization and establishment of the ubiquitin linkage SGA protocol

A detailed four-marker SK1 SGA protocol is outlined in *Figure 1—figure supplement 4*, and is modified from the two-marker S288C SGA protocol with the key optimization steps described below (*Schuldiner et al., 2006*). To ensure the quality of the genetic interaction screen, multiple optimizations were performed at various steps of the ubiquitin linkage SGA. During the pinning steps with a Singer ROTOR instrument, a mixed pinning setup was used to maximize the number of cells transferred. Linkage analysis was used in pilot screens to determine whether the selections were functional and recovered accurate genetic interactions. Specifically, since the SGA is selecting for haploid cells with modified ubiquitin loci, crossing all K-to-R ubiquitin strains to the *ubi2* deletion strain from the SK1 deletion library will result in lethality.

Sporulation of the SK1 strain is 90–100% efficient compared to the 10–16% sporulation efficiency of S288C strain (*Deutschbauer and Davis, 2005*). Conventional SGA studies using S288C sporulate cells for 5 days. The high sporulation efficiency of SK1 allowed sporulation for 2 days at 25°C, thereby shortening the duration of the genetic screen significantly.

It is essential to determine the optimal concentration of G418, as this selection is required to select against the wild-type allele of the K-to-R mutant strains and select for the specific gene deletion that comes from the library. Previous SGA studies have used between 100 mg/L and 200 mg/L G418 (*Collins et al., 2010*). Titration of various G418 concentrations for SK1 indicated that this strain requires higher G418 concentrations. 350 mg/L was used for the ubiquitin linkage SGA.

After sporulation, canavanine is used to kill the remaining diploids. In the S288C-SGA, since sporulation is inefficient, 50 mg/L Canavanine is included throughout the SGA workflow (following sporulation) (*Figure 1—figure supplement 4*) (*Geng et al., 2012*). In the SK1 SGA, including 50 mg/L Canavanine along with the mutant selection steps (e.g. 50 mg/L Canavanine +350 mg/L G418) resulted in sickness and changes in the colony sizes (data not shown). Titration of canavanine, in combination with the various selection drugs used in the SGA workflow, revealed that 10 mg/L of canavanine was sufficient to kill diploids without causing growth defects. 50 mg/L of canavanine was still used for haploid selection following sporulation, and 10 mg/L was included in all subsequent selection plates (*Figure 1—figure supplement 4*).

The K-to-R ubiquitin loci were selected prior to selecting for the deletion array, as linkage analysis with *ubi2* revealed that the reverse order did not efficiently select for cells carrying the modified ubiquitin loci. Haploid mutant selections were performed in the following order: (1) C-uracil plates (*ubi3*), 200 mg/L hygromycin (*ubi2*); (2) 100 mg/L Nourseothricin (*ubi4*); (3) 350 mg/L G418 (single gene deletions). In each drug selection step, the previous selection drug or condition was included to ensure precise mutant selections (*Figure 1—figure supplement 4*).

## SGA data analysis

The final double mutant selection plates were grown at 30°C for 48 hr, and then photographed. Colony sizes were recorded and normalized, and genetic interaction scores (S-scores) were calculated using established SGA protocols (*Collins et al., 2010*; *Collins et al., 2006*). Of the 2655 gene deletion strains used in the final analysis, 576 strains were analyzed based on three technical replicates, while the rest were analyzed based on two biological replicates, each with three technical replicates. The two biological replicates were used to compute the correlation coefficient between replicates of

the screen. Hierarchical clustering of mutants was performed using the Gene Cluster 3.0 software, and data was visualized with Java TreeView (https://sourceforge.net/projects/jtreeview). The Pearson correlation coefficient between the genetic interactions of the K-to-R ubiquitin mutants and the genetic interactions of thousands of single-gene deletions was calculated using a combined budding yeast genetic interactome data set (*Ryan et al., 2012*). Correlation coefficients calculated from genetic interaction profiles with 500 or more genetic interaction pairs are graphed in *Figure 3E*. For analysis of gene ontology term enrichment, the genetic interactions of K-to-R ubiquitin mutants with S-scores of $-2$ or less were analyzed for the enrichment of biological process (BP_Direct) GO terms using the Database for Annotation, Visualization and Integrated Discovery (DAVID) tool (*Huang et al., 2009a*; *Huang et al., 2009b*).

## Analysis of genetic similarity enrichment within physically interacting gene pairs

The degree of enrichment of genetic interaction profile similarity among gene pairs that physically interact was compared to gene pairs that are not known to physically interact. This comparison was carried out using the genetic interactions in the ubiquitin SGA, as well as the chromosome biology E-MAP (*Collins et al., 2007*). The analysis was done for the 392 gene deletions that were present in both screens. Physical interaction data was obtained from *Collins et al. (2007)*. A threshold of a PE score greater than two was used to determine gene pairs that physically interact. The Pearson correlation coefficient for all gene deletions was calculated based on the similarity of their genetic interaction profiles in each screen. The correlation coefficients of physically interacting and non-interacting genes were graphed.

## Strains, cell culture conditions, and plasmids

Plasmids and strains used in this study are listed in *Supplementary file 1* and *2*, respectively. All yeast strains in this study are in the SK1 background. Cells were cultured at 30°C on YM-1 media supplemented with 2% dextrose, unless otherwise noted. Selective media lacking specific amino acids or nucleobases was made using Complete Supplement Mixture (CSM) from Sunrise Sciences, supplemented with yeast nitrogen base and 2% glucose. Cloning of constructs and transformations were done using standard techniques.

## Validation of genetic interactions

K-to-R ubiquitin mutant strains (DS368-397, *Supplementary file 2*) were mated to desired single-gene deletions from the SK1 deletion library. Diploid cells were sporulated at 23°C, and tetrads were dissected on YM-1 plates with 2% dextrose. Spores were then genotyped by replica-plating onto appropriate plates, and the desired mutant spores were selected. Petite colonies were identified by lack of growth on plates containing 3% glycerol as the carbon source and discarded. To analyze the growth of strains, overnight cultures were normalized to an OD600 value of 1, then five-fold serial dilutions were spotted onto CSM plates supplemented with 2% glucose, unless otherwise indicated, grown for 3 days at 30°C and then scanned. It is worth noting that, as with all SGA analyses, a quantitative validation of genetic interactions requires the replication of the full SGA protocol since the presence of any or all of the four selectable markers used could affect the genetic interaction.

## Threonine uptake assay

Overnight cultures were diluted to OD600 of 0.05 in B-liquid, pH4.5, supplemented with 0.01 mg/mL histidine, 0.02 mg/mL uracil, and 0.1 mM threonine. When cultures reached OD600 of 0.3, 6 mL were collected for time point zero and placed on ice with 120 µL of 1M $NaN_3$. The cells were incubated on ice for the duration of the experiment, then 0.5uCi of 3–3 H L-threonine (American Radiolabeled Chemicals, ART 0330) were added to time point zero, and mixed by inverting once. Cells were then immediately collected by filtration with glass microfiber filters (Whatman, 934-AH), and then washed with 25 mL of cold water. The filter was then placed in a scintillation vial and dried overnight at 55°C. To begin the time-course experiment, 3uCi of 3–3 H L-threonine were added to each 34 mL culture. For each indicated time point, 6 mL were collected by filtration as described above. The OD600 of the cultures was recorded at each time point. Radioactivity was measured by liquid scintillation counting (Research Products International, Bio-Safe NA; Beckman, LS6500) for 5

min. Experiments were performed in triplicate. The slope was calculated by linear regression, and a two-sample T-test was performed to determine statistical significance.

## Quantitative proteomics

The indicated strains were grown to mid-log phase in C-Lys media containing either 13C6, 15N2 (Cambridge Isotope Laboratories, Tewksbury MA), or unlabeled lysine, both at 0.06 mg/mL. Sufficient incorporation of labeled lysine was attained by allowing cells grow for more than eight generations. Cells were collected by centrifugation, washed with cold HPLC-grade water and flash frozen. Cell pellets were then resuspended in lysis buffer (8M urea, 0.1M Tris-HCl pH 8, 150 mM NaCl, Roche mini protease inhibitor tablet without EDTA) and lysed by bead beating (20 1-min bursts with rests of 2 min following each cycle). Lysates were collected and incubated at room temperature with rotation. The lysates were then clarified by centrifugation at 17,000 x g for 10 min, followed by a second clarification cycle on the collected supernatant. The protein concentration in each sample was determined by BCA (Pierce) and samples were normalized accordingly. Disulfides were reduced with 4 mM TCEP and cysteines alkylated with 10 mM iodoacetamide, both for 30 min at room temperature in the dark. Excess iodoacetamide was quenched with 10 mM DTT for 30 min at room temperature in the dark. The samples were diluted to reduce the urea concentration to 2M with 0.1M Tris-HCl pH 8. Trypsin (in 50 mM acetic acid) was added at a ratio of 1 mg of enzyme to 100 mg of protein in the sample, and samples were incubated at room temperature overnight with end-to-end rotation.

After digestion, 10% TFA was added to a final concentration of 0.3–0.1% TFA, with pH of final solution <3. Insoluble material was precipitated by centrifugation and peptides were loaded onto a Sep Pak tC18 column that had been activated with 1 mL 80% ACN/0.1% TFA and equilibrated with 3 × 1 mL 0.1% TFA. Columns were washed with 5 × 1 mL 0.1 TFA and eluted with 1 mL 40% ACN/0.1% TFA prior to lyophilization. Diglycine-modified peptides were immunoprecipitated using an antibody specific for the K-GG motif characteristic of trypsinized, Ub-modified peptides (UbiScan, Cell Signaling Technology). Immunoprecipitates were then desalted using C18 Ultra MicroSpin columns (Nest Group), evaporated to dryness, and reconstituted in 0.1% formic acid for LC-MS/MS analysis.

Samples were analyzed on a Thermo Fisher Orbitrap Fusion mass spectrometry system equipped with a Easy nLC 1200 ultra-high pressure liquid chromatography system interfaced via a Nanospray Flex nanoelectrospray source. Samples were injected on a C18 reverse phase column (25 cm x 75 µm packed with ReprosilPur C18 AQ 1.9 µm particles). Peptides were separated by an organic gradient from 5% to 30% ACN in 0.1% formic acid over 120 min at a flow rate of 300 nL/min. The MS continuously acquired spectra in a data-dependent manner throughout the gradient, acquiring a full scan in the Orbitrap (at 120,000 resolution with an AGC target of 200,000 and a maximum injection time of 100 ms) followed by as many MS/MS scans as could be acquired on the most abundant ions in 3 s in the dual linear ion trap (rapid scan type with an intensity threshold of 5000, HCD collision energy of 29%, AGC target of 10,000, a maximum injection time of 35 ms, and an isolation width of 1.6 m/z). Singly and unassigned charge states were rejected. Dynamic exclusion was enabled with a repeat count of 1, an exclusion duration of 20 s, and an exclusion mass width of ±10 ppm.

Raw mass spectrometry data were assigned to *S. cerevisiae* protein sequences and MS1 intensities extracted with the MaxQuant software package (version 1.5.5.1) (*Cox and Mann, 2008*). Data were searched against the SwissProt *S. cerevisiae* protein database. Variable modifications were allowed for N-terminal protein acetylation, methionine oxidation, and lysine diglycine modification. A static modification was indicated for carbamidomethyl cysteine. The 'Match between runs' feature was enabled to match within 2 min between runs. All other settings were left using MaxQuant default settings.

## Cell cycle and half-life analysis experiments

Overnight cultures grown at room temperature were diluted to OD600 of 0.25, and then arrested with 15 µg/mL alpha factor at room temperature for 4 hr. More alpha factor was added every 2 hr. To release cells from arrest, cells were washed and resuspended in fresh media, then shifted to 30°C. Small samples were taken for flow cytometry analysis. Alpha factor was added to cultures 45 min after release to prevent cells from re-entering the cell cycle following mitosis. For experiments

analyzing the half-life of proteins, cycloheximide was added to cultures at 200 µg/mL at the indicated time points. For experiments analyzing the half-life of proteins in G1-arrested cultures, cells growing in YM-1 media with 2% raffinose were arrested in G1 phase as described above. Cells were then collected by centrifugation, washed and resuspended in YM-1 media containing 4% galactose and 15 µg/mL alpha factor for 1.5 hr, and shifted to 37°C. Cultures were then centrifuged and cells were resuspended in YM-1 containing 2% glucose, and 200 µg/mL cycloheximide. Cell pellets were collected by centrifugation at the indicated time points, washed with cold water, and then flash frozen. For flow cytometry analysis, cells were fixed in 70% ethanol, then treated at 50°C with RNase A (0.25 mg/mL in 50 mM sodium citrate) and Proteinase K (0.063 mg/mL in 50 mM sodium citrate) for 1 hr each, followed by incubation with Sytox Green (1:10,000) overnight at 4°C.

## Western blots and antibodies

Cell pellets were resuspended in 100 µl of Sina's loading buffer (50 mM Tris pH7.5, 5 mM EDTA, 10% glycerol, 5% SDS, 0.5% 2-mercaptoethanol, 0.1% w/v bromophenol blue) and 50 µl of glass beads. Samples were incubated at 100°C for 7 min, then processed in bead beater from 3 min, followed by heating at 100°C for 7 min. Cell lysates were analyzed by SDS-PAGE using 4–20% gradient Tis-HCl gels (BioRad, #3450034). Proteins were transferred onto 0.2-µm nitrocellulose membrane. Membranes were blocked with 5% nonfat dry milk in PBS-T or Odyssey Blocking Buffer. Rabbit antibodies recognizing Clb2, Clb3, Clb5, and Pds1 were generous gifts from Adam Rudner. Cdc28 was analyzed with goat antibody from Santa Cruz Biotechnology (yC-20). Mouse anti-PSTAIR (P7962) and Flag epitope (F3165) were purchased from Millipore-Sigma. Rabbit anti-ubiquitin antibody was purchased from Thermo-Fisher (PA3-16717) or Santa Cruz Biotechnology (sc-9133).

## In vitro APC ubiquitination assay

APC was immunopurified using rabbit IgG beads (Invitrogen 14301) from a yeast strain expressing Cdc16-TAP::HIS3 and lacking Cdh1 (*cdh1*::LEU2) in lysis buffer (25 mM HEPES 7.5, 150 mM KOAc, 5 mM MgCl2, 10% Glycerol, 0.2% Triton X, 1 mM DTT, 1 mM PMSF) for 1.5 hr at 4°C following bead beating. Beads were washed 3X with 25 mM HEPES 7.5, 150 mM KOAc, 10% Glycerol, 0.1% Triton X.

Cdh1 was in vitro translated using Promega kit (Promega L1170). Cdh1 was then incubated with the APC on beads from the above step for 30 min at room temperature. Beads were washed 3X with 25 mM HEPES 7.5, 150 mM KOAc, 10% Glycerol, 0.1% Triton X.

Hsl1 (667-872)-TAP was in vitro translated using Promega kit (Promega L1170) using 35S-Methionine. After in vitro translation, the Hsl1 fragment was incubated with IgG-coupled magnetic beads for 30 min, and washed 3X with 25 mM HEPES 7.5, 150 mM KOAc, 10% Glycerol, 0.1% Triton X. Hsl1 was cleaved off the beads with TEV protease for 30 min (AcTEV - Thermo Fisher 12575015).

Wild type (U-100Sc), K11-only (UM-K110) and K0 (UM-KOK) ubiquitin was purchased from Boston Biochem, INC. Uba1 and Ubc4 were expressed in *E. coli* and purified. Ubc4 was charged with ubiquitin using 0.2 mg/mL Uba1, 2 mg/mL Ubc4, 2 mg/mL ATP, and 2 mg/mL ubiquitin. The reaction was incubated at 37°C for at least 30 min.

APC-Cdh1 was incubated with charged Ubc4 and TEV purified Hsl1 fragment at 25°C for 45 min. Reactions were terminated by adding 2X SDS loading dye and analyzed by SDS-PAGE.

## Additional information

### Competing interests

Deniz Simsek: is affiliated with Amgen Research. The author has no other competing interests to declare. The other authors declare that no competing interests exist.

### Funding

| Funder | Grant reference number | Author |
| --- | --- | --- |
| National Institutes of Health | R35 GM118104 | David P Toczyski |

The funders had no role in study design, data collection and interpretation, or the decision to submit the work for publication.

### Author contributions
Fernando Meza Gutierrez, Conceptualization, Data curation, Formal analysis, Validation, Investigation, Visualization, Methodology, Writing—original draft, Writing—review and editing; Deniz Simsek, Conceptualization, Data curation, Formal analysis, Validation, Investigation, Visualization, Methodology, Writing—review and editing; Arda Mizrak, Validation, Investigation, Methodology; Adam Deutschbauer, Michelle Nguyen, Resources, Validation; Hannes Braberg, Formal analysis, Validation, Visualization, Methodology; Jeffrey Johnson, Formal analysis, Validation, Investigation, Methodology; Jiewei Xu, Investigation, Methodology; Michael Shales, Data curation, Formal analysis, Methodology; Raquel Tamse-Kuehn, Curt Palm, Lars M Steinmetz, Resources, Investigation, Methodology; Nevan J Krogan, Conceptualization, Supervision, Methodology, Project administration; David P Toczyski, Conceptualization, Supervision, Funding acquisition, Investigation, Methodology, Project administration, Writing—review and editing

### Author ORCIDs
Fernando Meza Gutierrez (iD) http://orcid.org/0000-0002-5601-7202
David P Toczyski (iD) http://orcid.org/0000-0001-5924-0365

### Decision letter and Author response
Decision letter https://doi.org/10.7554/eLife.42955.021
Author response https://doi.org/10.7554/eLife.42955.022

## Additional files
### Supplementary files
• Supplementary file 1. Plasmids used in this study.
DOI: https://doi.org/10.7554/eLife.42955.014

• Supplementary file 2. Strains used in this study.
DOI: https://doi.org/10.7554/eLife.42955.015

• Supplementary file 3. Oligonucleotides used in this study.
DOI: https://doi.org/10.7554/eLife.42955.016

• Supplementary file 4. The full Ubiquitin SGA dataset.
DOI: https://doi.org/10.7554/eLife.42955.017

• Supplementary file 5. Quantitative proteomics analysis of the ubiquitinome in $doa10\Delta$, $ubc6\Delta$, $ubc7\Delta$, $ubc6,7\Delta$.
DOI: https://doi.org/10.7554/eLife.42955.018

• Transparent reporting form
DOI: https://doi.org/10.7554/eLife.42955.019

### Data availability
All datasets generated are included as Supplementary Files 4 (SGA dataset) and 5 (quantitative proteomics).

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
