## [Decision Letter]

[Editors’ note: a previous version of this study was rejected after peer review, but the authors submitted for reconsideration. The first decision letter after peer review is shown below.]

Thank you for submitting your work entitled "Genetic Analysis Reveals Functions of Atypical Polyubiquitin Chains" for consideration by *eLife*. Your article has been reviewed by three peer reviewers, one of whom is a member of our Board of Reviewing Editors, and the evaluation has been overseen by a Senior Editor. The reviewers have opted to remain anonymous.

Our decision has been reached after consultation between the reviewers. Based on these discussions and the individual reviews below, we regret to inform you that your work will not be considered further for publication in *eLife*.

While all reviewers noted the strength of the genetic screen you employed for discovering new functions of specific ubiquitin linkages, they also noted shortcomings. There was an agreement among the reviewers, particularly after careful discussion of both the paper and the reviews, that in its current state it is not clear whether the K11 function for threonine import is due to substrate modification with K11-linked chains or due to a distinct role of Lys11 of ubiquitin (as had been observed with other Lys residues that fulfill binding or recognition functions). This concern is in part due to a lack of biochemical follow-up of your genetic discoveries, which the reviewers considered to be essential. Finally, there was some concern about novelty of your findings in light of lack of more detailed mechanistic insight. As the experiments required by the reviewers (again after careful discussion among all reviewers) would take much longer than the 2-3 months usually accepted by *eLife*, we believe that your paper will find a better home in a different journal.

*Reviewer #1:*

The functional consequences of modification with ubiquitin chains of distinct topologies in cells remain ill-defined. To address this shortcoming, the authors of this paper developed a powerful synthetic genetic interaction platform testing for genetic interactions of six out of seven possible Lys-Arg ubiquitin modifications in yeast. As ubiquitin is encoded by four genes, this required the development of a new SGA library in SK1 yeast strains, an effort that will allow more complex genetic interaction studies to be conducted in the future. The results of their SGA screen provides a rich data-source with thousands of candidate interactions between gene deletions and specific ubiquitin Lys-Arg mutations; thus, these findings will be very interesting for a broad readership and could be a starting point for many new investigations. They validated two genetic interactions of K11R mutant yeast with enzymes of threonine biosynthesis and cell cycle regulation; in particular, their work on K11-ubiquitin dependent regulation of threonine import provides a new angle for K11-linked ubiquitin chains and thus underscores the potential of the chosen genetic approach. However, the authors were unable to provide mechanistic insight into the functions of K11 linkages in controlling threonine import, as the substrates as well as the nature of the modification (homotypic K11-linked chain versus more complex mixed or branched chains in which K11-linkages constitute only one out of several linkages) remain unknown.

Together, I am convinced that this manuscript provides rich data for many researchers that are interested in understanding functional consequences of ubiquitin chain modifications in cells. Given the lack of mechanistic insight, I would argue that this paper should be published as a Tools and Resources article, rather than a Research Article, after the following issues have been addressed.

1) Given that the SGA analysis is the main strength of the paper and will generate broad interest, I believe it is pivotal that the data can be interrogated in an easy, online manner. This could be achieved by providing a searchable database for genetic interactions observed in this study.

2) The stabilization phenotypes in Figure 4 are weak, i.e. as they are only observed in cells lacking CDC26. To provide a more solid foundation for these results, the degradation experiments in Figure 4B, C need to be repeated, quantified, and analyzed with appropriate statistical tools.

3) It is interesting that K11R mutants do not drive clustering, as opposed to other K-R mutants. A potential explanation for this could be that in yeast, K11-linkages are predominantly used for chain initiation, allowing for branching off of chain blocks that in turn determine the outcome of the modification. This would be consistent with their two examples (particularly with the APC/C, where K48-linkages are thought to provide the main degradation signal in yeast). The authors might want to discuss this possibility more explicitly.

*Reviewer #2:*

Protein ubiquitination plays a critical role in almost all cellular events. Diverse polyUb chains have been found to be commonly present in cells, and harbor different dynamic structures, although their abundance appears to be different, dependent on the total number and absolute levels of modified substrates. The Ub chains have been recognized to contribute to signaling specificity in Ub conjugation, recognition, de-conjugation, and functional consequences. Dissection of the function of these diverse polyUb modifications, however, is still technically challenging and largely limited to the studies of individual substrates. Meza-Gutierrez et al. attempted to explore physiological functions of individual polyUb chains by large-scale genetic interactions. Although the scope of the study was large, and the link of K11 chains to amino acid homeostasis and cell cycle regulation is reliable, there are some concerns with unknown false discovery rate, indirect consequence of genetic mutations, and lacking of solid biochemical analysis. Thus, as in current status, the manuscript may not be suitable for the publication in *eLife*, but is more appropriate for a specialized journal in genetics. Specific concerns are as follows:

1) In addition to 7 lysine residues, the N-terminal amine group is also involved in the assembly of polyUb chain, in particular in some immunological pathways in mammalian cells. Although N-terminal polyUb is not essential in yeast, shown in Dr. Dan Finely's genetics analysis with N-terminal tagged Ub, it may still be functional in specific pathways. This point is not mentioned in the Introduction, and is not considered during the genetics interaction assay.

2) As Ub K48 is essential for yeast growth, the authors generated a strain with K48R but maintaining 33% expression of wild type ubiquitin. The rationale for 33% is not well specified. What is the minimal level of wild type Ub required for the assay. In addition, K63R mutation was not used due to another technical issue.

3) It may be discussed that only non-essential gene deletions (n = 2,655) were used in this study. Another ~50% genes were not tested. It is not known if the 2,655 genes were expressed and well represented in the ubiquitinated proteome. Since multiple large-scale ubiquitinated studies were reported in yeast, it may be useful to examine whether the 2,655 proteins are ubiquitinated and unbiasedly detected in the ubiquitinated proteome.

4) In the last paragraph of the subsection “The ubiquitin linkage synthetic genetic array”, a small group of genetics interactions were validated in an independent analysis. It is important to estimate the false discovery rate of the initial genetic interaction analysis at the global level.

5) Subsequent genetic studies of K11 chains involved in amino acid homeostasis and mitotic progression are interesting and comprehensive, including the usage of numerous Hom2, Hom3 and Doa10 mutations, although the function of K11 in APC modification is well known in mammalian cells. The Thr pathway discovery is novel but not fully investigated by other complementary tools, such as biochemical or mass spectrometric approaches to identify potential targets, cell biology approaches to investigate potential protein distribution (sorting or internalization) defect in K11R mutation.

6) In addition, to make the discovery more appealing, it is ideal to address whether the K11 linkage functions to regulate amino acid homeostasis in mammalian cells.

*Reviewer #3:*

The manuscript by Meza-Gutierrez et al., describes an SGA screening strategy to uncover potential functions of specific ubiquitin linkages in the budding yeast. Using the output of their screen, the authors drill down on two of the promising genetic interactions uncovered, specifically the potential role of Lysine-11 linkages in threonine biosynthesis and in APC substrate turnover. While the experimental results overall validate that the discovered genetic links are real (not simply noise), the novelty of the discoveries and the lack of understanding of the underlying mechanism of action tempers this reviewer's enthusiasm.

The authors generated yeast haploids with individual and combinations of lysine-to-arginine mutations in each of three ubiquitin genes (plus deletion of the Ubl1 locus) and crossed/mated these to a haploid gene deletion array (also containing the Ubl1 deletion). Selection and high sporulation efficiency to generate haploids containing the ubiquitin mutation and non-essential gene deletion was accomplished in the SK1 yeast strain to reduce background from unsporulated diploid cells.

Growth of mutant strains was measured and clustered to identify genetic interactions. Known gene modules were identified by the clustering approach providing confidence in the process. (Figure 1B). A subset of hits (*ubc4Δ, hom2Δ* and *cdc26Δ*) were characterized in more detail (Figure 2) to demonstrate that the SGA screen results are reproducible and robust.

Two genetic interactions with the K11-Ubiquitin mutant were then chosen for further investigation, in part because K11-linkages are prevalent in yeast and their functional significance is poorly understood, and because of the proven importance of K11-linkages in metazoan biology.

First: The discovery of synthetic growth defects for *HOM2* and *HOM3* genes within the homoserine biosynthesis pathways suggested functional redundancy of K11-Ubiquitin linkages with amino acid homeostasis (Figure 3A-C). In support of this hypothesis, the growth defect of *K11/hom2* and *K11/hom3* double mutant strains were rescued by supplementation of homoserine or threonine but not methionine (Figure 3D).

The genetic interactome displayed by the K11R Ub linkage mutant was suggestively similar to that uncovered previously for *GNP1*, a threonine permease. To assess if K11-ubiquitin linkages serve a similar role to *GNP1* in threonine import, the authors measured uptake of radioactive threonine for WT and K11R mutant strains in a time course experiment. In support of a role for K11 chains in Thr transport, the K11R mutant displayed almost a 2 fold lower transport (Figure 3F). Note that it would have been useful to show the relative effect of the *gnp1Δ* mutant for comparison.

Interestingly, the E3 ligase Doa10 responsible for the majority of K11-linked ubiquitin chains in yeast also showed a synthetic growth defect that could be rescued by homoserine (Figure 3G). This discovery is very important because it provides the first (and only?) corroborating piece of evidence to suggest that the loss of K11 ubiquitin chain linkages may actually underpin the detected genetic interaction. Up to this point in the manuscript, the authors have not ruled out the possibility (and still it remains possible) that the genetic interactions detected relate to the ability of the K11R mutation to affect some function of Ub beyond its ability to form K11 chains. For example, the K11 side chain of ubiquitin could be involved in a physiological protein-protein interaction that could not be faithfully mimicked by arginine.

This raises a major criticism of the paper – namely that until the full or minimally a major portion of the underlying mechanism of action of the K11R mutation for the process in question is figured out, there is no certainty that the working hypothesis of a role for specific chain linkage is correct.

Second: The authors then investigate the strong genetic interaction of the K11R Ub mutant with CDC26, a non-essential subunit of the APC. In Figure 4, the authors show that levels of a number of known APC substrates are affected by the K11R mutation including Clb1 and Pds1/securin most strongly and Clb3, and Clb5 to a lesser extent. The key question that remains to be addressed is whether the effects of the K11R mutation on these substrate levels are direct or indirect. Until the mechanism of the K11R mutation is fleshed out in more detail, one cannot say with any certainty. Since the role of K11 chains is well established for the metazoan APC, this fact lends some support to the notion that the K11 Ub linkage type could play a direct role in APC function in yeast. However, this point also has the effect of subtracting from the novelty of their discovery. The fact that K11R mutant has no effect on WT yeast also suggests that K11 linkages are not all that important to yeast biology.

Additional points:

1) It would be informative if the authors were to show FACS profiles for the analyses described in Figure 4.

2) The result of the Figure 4B would be more compelling if the authors were to include quantification of band intensities in Figure 4. Also include additional loading controls.

[Editors’ note: what now follows is the decision letter after the authors submitted for further consideration.]

Congratulations, we are pleased to inform you that your article, "Genetic Analysis Reveals Functions of Atypical Polyubiquitin Chains", has been accepted for publication in *eLife*.

This manuscript introduces an important genetic approach to dissect signaling through distinct ubiquitin chain topologies. It has used synthetic interaction screens to reveal novel functions of K11-linked chains in threonine important, as well as to provide an interesting new angle on the role of K11-linked chains in yeast cell cycle regulation. The technology is important and the screen results provide interesting starting points for many new projects, and hence, this manuscript should be distributed widely. However, the molecular insights coming out of the screen are still a bit less well developed, and this paper has to be published in the "Tools and Resources" format.

*Reviewer #1:*

The authors of this resubmitted manuscript addressed some of the initial concerns raised by all three reviewers. However, they did not identify a direct substrate for K11-specific ubiquitylation that plays a role in threonine import. The identification of K11-linkages in ubiquitin chains attached to APC-substrates in yeast has been validated biochemically, yet the architecture of the resulting conjugates has also not been addressed (as would have been expected given the known chain architecture of human APC). As I had mentioned during the first round of review, I therefore believe that this manuscript is worthwhile to be published in *eLife* as it provides a novel way to identify cellular functions of particular ubiquitin linkages; however, given the lack of molecular insight, I would argue this should be published as a Tools and Resources article, and not as a Research Article.

*Reviewer #2:*

All of my comments have been well addressed by additional experiments and discussion. The revised manuscript is significantly improved and acceptable for publication.

---

## [Author Response]

[Editors’ note: the author responses to the first round of peer review follow.]

While all reviewers noted the strength of the genetic screen you employed for discovering new functions of specific ubiquitin linkages, they also noted shortcomings. There was an agreement among the reviewers, particularly after careful discussion of both the paper and the reviews, that in its current state it is not clear whether the K11 function for threonine import is due to substrate modification with K11-linked chains or due to a distinct role of Lys11 of ubiquitin (as had been observed with other Lys residues that fulfill binding or recognition functions). This concern is in part due to a lack of biochemical follow-up of your genetic discoveries, which the reviewers considered to be essential. Finally, there was some concern about novelty of your findings in light of lack of more detailed mechanistic insight. As the experiments required by the reviewers (again after careful discussion among all reviewers) would take much longer than the 2-3 months usually accepted by eLife, we believe that your paper will find a better home in a different journal.

All reviewers coincided in their major criticism, namely a lack of biochemistry, leaving open the possibility that the amino acid import and APC-substrate turnover defects observed in K11R ubiquitin mutant cells are unrelated specifically to K11-linked polyubiquitin chains. As noted by reviewer 3, data showing that *doa10Δ* phenocopies the K11R result related to amino acid import “is very important because it provides the first (and only?) corroborating piece of evidence to suggest that the loss of K11 ubiquitin chain linkages may actually underpin the detected genetic interaction.” However, they note that our discovery of a “strong genetic interaction of the K11R Ub mutant with CDC26” and that “that levels of a number of known APC substrates are affected by the K11R mutation” could be indirect because “The key question that remains to be addressed is whether the effects of the K11R mutation on these substrate levels are direct or indirect. Until the mechanism of the K11R mutation is fleshed out in more detail, one cannot say with any certainty.” We took these criticisms to heart. In the case of *DOA10*, we performed SILAC mass spectrometry analysis, but were unable to identify substrates likely responsible for the phenotype (included in the supplement). However, as noted by the reviewer, the identification of *DOA10* already provides significant corroboration. We therefore concentrated on our APC data.

We now show that the yeast APC and its cognate E2 enzyme, Ubc4, modify substrates with K11- linked polyubiquitin chains in vitro. This was carried out not only with a WT fragment of Hsl1, which has many lysine acceptors, but also a novel Hsl1 variant with only a single target lysine. These results clearly show that the E2 that generates base chains in yeast, Ubc4, adds K11 linked chains. When viewed in combination with direct alterations of substrate turnover rates in vivo and the synthetic growth phenotypes we reported in our original submission to *eLife*, our results provide strong and direct in vivo and in vitro evidence that polyubiquitination of APC substrates with K11-linked ubiquitin chains promotes their timely degradation, thereby contributing to normal cell cycle progression.

Importantly, our in vitro and molecular data suggest that, while K11 and K48 are the two critical linkages in both yeast and humans, the architecture of ubiquitin chains on APC substrates differs. In humans, K48 is thought to exist in the base chain, and these are extended with homogenous K11, whereas our data are consistent with yeast APC having a critical K11 linkage in the base chain, with a more homogenous K48 addition onto that. Thus, we have uncovered a related, but distinct role for this linkage in APC biology. This provides an interesting perspective on the evolutionary conservation of the APC: The linkages important for APC function are highly conserved, but the architecture in which they are employed is not.

Two reviewers of our manuscript requested a quantitative analysis of the representative western blot images presented in Figure 4, which show the delayed degradation of various APC substrates observed in K11R ubiquitin mutant cells. To address this criticism, we have now undertaken extensive efforts to quantify and normalize band intensities, and use these data to present precise half-lives in Figure 4.

We would like to point out that there is a long tradition in molecular biology of studying posttranslational modifications by examining mutants that eliminate phosphorylation or acetylation sites. While there will always be some chance that a single conservative mutation will cause a non-specific effect, this type of analysis is the very backbone of the kinase and HAT fields, and has provided tremendous insights into these fields. K11 has been shown to be present and abundant in vivo; we have found growth phenotypes associated with specific mutations; we have shown that the an E3 (Doa10) thought to generate such linkages has a similar phenotype; we have shown that APC activity is compromised in vivo; and we have now shown that the APC generates such linkages in vitro. We feel that this provides enormous support for our ability to identify functional roles of these linkages in vivo.

Reviewer #1:The functional consequences of modification with ubiquitin chains of distinct topologies in cells remain ill-defined. To address this shortcoming, the authors of this paper developed a powerful synthetic genetic interaction platform testing for genetic interactions of six out of seven possible Lys-Arg ubiquitin modifications in yeast. As ubiquitin is encoded by four genes, this required the development of a new SGA library in SK1 yeast strains, an effort that will allow more complex genetic interaction studies to be conducted in the future. The results of their SGA screen provides a rich data-source with thousands of candidate interactions between gene deletions and specific ubiquitin Lys-Arg mutations; thus, these findings will be very interesting for a broad readership and could be a starting point for many new investigations. They validated two genetic interactions of K11R mutant yeast with enzymes of threonine biosynthesis and cell cycle regulation; in particular, their work on K11-ubiquitin dependent regulation of threonine import provides a new angle for K11-linked ubiquitin chains and thus underscores the potential of the chosen genetic approach. However, the authors were unable to provide mechanistic insight into the functions of K11 linkages in controlling threonine import, as the substrates as well as the nature of the modification (homotypic K11-linked chain versus more complex mixed or branched chains in which K11-linkages constitute only one out of several linkages) remain unknown.Together, I am convinced that this manuscript provides rich data for many researchers that are interested in understanding functional consequences of ubiquitin chain modifications in cells. Given the lack of mechanistic insight, I would argue that this paper should be published as a Tools and Resources article, rather than a Research Article, after the following issues have been addressed.

We thank this reviewer for their interest in our manuscript. We have now made several modifications to quantify the data, as suggested. In addition, we have provided additional follow-up, which we hope the reviewer agrees provides more mechanistic insight. We should note that while we have maintained the manuscript as a Research Article, we would consider the Tools and Resources format, as suggested by the reviewer, if the editors feel this is the required.

1) Given that the SGA analysis is the main strength of the paper and will generate broad interest, I believe it is pivotal that the data can be interrogated in an easy, online manner. This could be achieved by providing a searchable database for genetic interactions observed in this study.

Supplementary file 4 includes the full ubiquitin SGA dataset formatted to be easily viewable using freely-available software tools. In particular, the authors recommend viewing the ubiquitin SGA as a heat map using Java TreeView, a free and user-friendly software that is compatible with the Windows, Macintosh, and Linux operating systems. Java TreeView can be downloaded from https://sourceforge.net/projects/jtreeview. This information is now provided in the manuscript.

2) The stabilization phenotypes in Figure 4 are weak, i.e. as they are only observed in cells lacking CDC26. To provide a more solid foundation for these results, the degradation experiments in Figure 4B, C need to be repeated, quantified, and analyzed with appropriate statistical tools.

As discussed above, we now report normalized, quantified band intensities in Figure 4.

3) It is interesting that K11R mutants do not drive clustering, as opposed to other K-R mutants. A potential explanation for this could be that in yeast, K11-linkages are predominantly used for chain initiation, allowing for branching off of chain blocks that in turn determine the outcome of the modification. This would be consistent with their two examples (particularly with the APC/C, where K48-linkages are thought to provide the main degradation signal in yeast). The authors might want to discuss this possibility more explicitly.

This is an interesting suggestion. In fact, our work suggests that to mediate APC-substrate degradation, K11 linkages function as chain initiators or anchors that are extended by long K48 chains, in contrast to the human APC. We now discuss this and include a model in Figure 4.

Reviewer #2:Protein ubiquitination plays a critical role in almost all cellular events. Diverse polyUb chains have been found to be commonly present in cells, and harbor different dynamic structures, although their abundance appears to be different, dependent on the total number and absolute levels of modified substrates. The Ub chains have been recognized to contribute to signaling specificity in Ub conjugation, recognition, de-conjugation, and functional consequences. Dissection of the function of these diverse polyUb modifications, however, is still technically challenging and largely limited to the studies of individual substrates. Meza-Gutierrez et al. attempted to explore physiological functions of individual polyUb chains by large-scale genetic interactions. Although the scope of the study was large, and the link of K11 chains to amino acid homeostasis and cell cycle regulation is reliable, there are some concerns with unknown false discovery rate, indirect consequence of genetic mutations, and lacking of solid biochemical analysis. Thus, as in current status, the manuscript may not be suitable for the publication in eLife, but is more appropriate for a specialized journal in genetics. Specific concerns are as follows:

We have performed additional analyses to evaluate the robustness of the genetic interaction data, and have now included a biochemical analysis of K11 linkages and the APC, which we hope the reviewer feels strengthens the manuscript.

1) In addition to 7 lysine residues, the N-terminal amine group is also involved in the assembly of polyUb chain, in particular in some immunological pathways in mammalian cells. Although N-terminal polyUb is not essential in yeast, shown in Dr. Dan Finely's genetics analysis with N-terminal tagged Ub, it may still be functional in specific pathways. This point is not mentioned in the Introduction, and is not considered during the genetics interaction assay.

We thank the reviewer for pointing out this omission. We now mention this in the Introduction.

2) As Ub K48 is essential for yeast growth, the authors generated a strain with K48R but maintaining 33% expression of wild type ubiquitin. The rationale for 33% is not well specified. What is the minimal level of wild type Ub required for the assay. In addition, K63R mutation was not used due to another technical issue.

While the exact minimum level of wild type ubiquitin required to carry out the SGA analysis, or to permit *S. cerevisiae* cells to survive was not determined, the K48R ubiquitin mutants in our study, aside from genetic interaction data, provide interesting insights into ubiquitin biology. The classic model for proteasomal degradation for ubiquitinated proteins has held that the minimum signal for efficient proteasomal degradation is a K48-linked chain of at least 4 ubiquitins. Because the strains used in the SGA analysis express only 20% wild type ubiquitin, the high abundance of K48R ubiquitin likely results in frequent early termination of chains generated by K48-specific E2 and E3 enzymes. (Note, there was a miscalculation in the original manuscript, the K48 actually represents 80, not 66, percent of the ubiquitin.) Although the lack of obvious phenotypes related to widespread protein degradation defects is surprising, it is consistent with reports by Braten et al. (2016), Martinez-Fonts et al. (2016), and others, which have suggested that monoubiquitin as well as non-48 or K11-linked chains can promote proteasomal degradation in some cases.

3) It may be discussed that only non-essential gene deletions (n = 2,655) were used in this study. Another ~50% genes were not tested. It is not known if the 2,655 genes were expressed and well represented in the ubiquitinated proteome. Since multiple large-scale ubiquitinated studies were reported in yeast, it may be useful to examine whether the 2,655 proteins are ubiquitinated and unbiasedly detected in the ubiquitinated proteome.

Proteomics studies are continually increasing the number of known ubiquitination sites. However, the precise proportion of proteins that are modified by ubiquitin in yeast is unknown. Nonetheless, based on published datasets curated by the *Saccharomyces cerevisiae* Genome Database (SGD), approximately a third of the genes in the ubiquitin SGA have been reported as being ubiquitinated.

4) In the last paragraph of the subsection “The ubiquitin linkage synthetic genetic array”, a small group of genetics interactions were validated in an independent analysis. It is important to estimate the false discovery rate of the initial genetic interaction analysis at the global level.

We thank the reviewer for pointing out this omission. The ubiquitin SGA had a correlation coefficient between replicates that is similar to what has been observed with other large genetic interaction screens. We have included a scatter plot showing the correlation between replicates of the screen in Figure 1C. The genetic interactions shown in Figure 3 provide a striking example of the robustness of the SGA dataset. With spot dilution assays, we are able to detect strong genetic interactions identified by the screen, such as the synthetic interaction between *HOM2/3* and K11 mutants, and several others shown in Figure 3. Significantly, weak genetic interactions are also reproducible, such as the mild synthetic interaction between *CDC26* and K11 and K33 ubiquitin mutants. In addition, we have now carried out an analysis of the correlation between known physical interactions and genetic interactions in our screen. Similar to what is observed with other genetic screens, such as the chromosome biology E-MAP (Collins et al., 2007), we find that genes known to physically interact are more likely to have common genetic interactions in our screen, and thus their genetic interaction profiles are more highly correlated than those for genes that do not interact physically (see Figure 1D). These observations underscore the robustness of the ubiquitin SGA.

5) Subsequent genetic studies of K11 chains involved in amino acid homeostasis and mitotic progression are interesting and comprehensive, including the usage of numerous Hom2, Hom3 and Doa10 mutations, although the function of K11 in APC modification is well known in mammalian cells. The Thr pathway discovery is novel but not fully investigated by other complementary tools, such as biochemical or mass spectrometric approaches to identify potential targets, cell biology approaches to investigate potential protein distribution (sorting or internalization) defect in K11R mutation.

We appreciate this point and undertook significant effort on this front. Unpublished results show that the expression and distribution of Gnp1, a threonine transporter, are undisturbed by mutation of ubiquitin’s K11. We have also undertaken a SILAC experiment in *doa10*∆ versus WT cells to identify substrates that might be responsible for this phenotype, but likely candidates were not uncovered. However, we have characterized the role of K11 in the APC in some detail. Importantly, while it is related to its role in human cells, it appears distinct in that it suggests the use of K11 as a base chain of the reaction.

6) In addition, to make the discovery more appealing, it is ideal to address whether the K11 linkage functions to regulate amino acid homeostasis in mammalian cells.

While this suggestion by the reviewer 2 would certainly be a fascinating extension of our results, it is outside the scope of the present work.

Reviewer #3:The manuscript by Meza-Gutierrez et al., describes an SGA screening strategy to uncover potential functions of specific ubiquitin linkages in the budding yeast. Using the output of their screen, the authors drill down on two of the promising genetic interactions uncovered, specifically the potential role of Lysine-11 linkages in threonine biosynthesis and in APC substrate turnover. While the experimental results overall validate that the discovered genetic links are real (not simply noise), the novelty of the discoveries and the lack of understanding of the underlying mechanism of action tempers this reviewer's enthusiasm.The authors generated yeast haploids with individual and combinations of lysine-to-arginine mutations in each of three ubiquitin genes (plus deletion of the Ubl1 locus) and crossed/mated these to a haploid gene deletion array (also containing the Ubl1 deletion). Selection and high sporulation efficiency to generate haploids containing the ubiquitin mutation and non-essential gene deletion was accomplished in the SK1 yeast strain to reduce background from unsporulated diploid cells.

*Growth of mutant strains was measured and clustered to identify genetic interactions. Known gene modules were identified by the clustering approach providing confidence in the process. (Figure 1B). A subset of hits (ubc4Δ*, *hom2Δ and cdc26Δ) were characterized in more detail (Figure 2) to demonstrate that the SGA screen results are reproducible and robust. Two genetic interactions with the K11-Ubiquitin mutant were then chosen for further investigation, in part because K11-linkages are prevalent in yeast and their functional significance is poorly understood, and because of the proven importance of K11-linkages in metazoan biology. First: The discovery of synthetic growth defects for HOM2 and HOM3 genes within the homoserine biosynthesis pathways suggested functional redundancy of K11-Ubiquitin linkages with amino acid homeostasis (Figure 3ABC). In support of this hypothesis, the growth defect of K11/Hom2 and K11/Hom3 double mutant strains were rescued by supplementation of homoserine or threonine but not methionine (Figure 3D). The genetic interactome displayed by the K11R Ub linkage mutant was suggestively similar to that uncovered previously for GNP1, a threonine permease. To assess if K11-ubiquitin linkages serve a similar role to GNP1 in threonine import, the authors measured uptake of radioactive threonine for WT and K11R mutant strains in a time course experiment. In support of a role for K11 chains in Thr transport, the K11R mutant displayed almost a 2 fold lower transport (Figure 3F). Note that it would have been useful to show the relative effect of the gnp1Δ mutant for comparison. Interestingly, the E3 ligase Doa10 responsible for the majority of K11-linked ubiquitin chains in yeast also showed a synthetic growth defect that could be rescued by homoserine (Figure 3G). This discovery is very important because it provides the first (and only?) corroborating piece of evidence to suggest that the loss of K11 ubiquitin chain linkages may actually underpin the detected genetic interaction. Up to this point in the manuscript, the authors have not ruled out the possibility (and still it remains possible) that the genetic interactions detected relate to the ability of the K11R mutation to affect some function of Ub beyond its ability to form K11 chains. For example, the K11 side chain of ubiquitin could be involved in a physiological protein-protein interaction that could not be faithfully mimicked by arginine. This raises a major criticism of the paper – namely that until the full or minimally a major portion of the underlying mechanism of action of the K11R mutation for the process in question is figured out, there is no certainty that the working hypothesis of a role for specific chain linkage is correct. Second: The authors then investigate the strong genetic interaction of the K11R Ub mutant with CDC26, a non-essential subunit of the APC. In Figure 4, the authors show that levels of a number of known APC substrates are affected by the K11R mutation including Clb1 and Pds1/securin most strongly and Clb3, and Clb5 to a lesser extent. The key question that remains to be addressed is whether the effects of the K11R mutation on these substrate levels are direct or indirect. Until the mechanism of the K11R mutation is fleshed out in more detail, one cannot say with any certainty. Since the role of K11 chains is well established for the metazoan APC, this fact lends some support to the notion that the K11 Ub linkage type could play a direct role in APC function in yeast. However, this point also has the effect of subtracting from the novelty of their discovery. The fact that K11R mutant has no effect on WT yeast also suggests that K11 linkages are not all that important to yeast biology.*

We thank the reviewer for their interest in the genetic interactions in our study. The reviewer feels that two factors detract from our follow-up analysis of the genetic interaction between K11R and an APC mutant: first, the lack of biochemical corroboration that K11 is used in yeast; second, the fact that K11 has been implicated in human APC biology, which the reviewer feels detracts from the novelty of the story. In addressing the first point, we have done further characterization of the yeast APC in vitro, which we hope the reviewer will agree deepens our mechanistic understanding of the function of K11 linkages. Specifically, we have shown that the APC, in conjunction with the initiating E2 in yeast, Ubc4, adds K11 chains to a model substrate. Thus, we have now shown that K11 is used in vitro, K11 alters APC activity in vivo, and used genetics to show that this alteration has clear biological effects on viability. As for the second point, our analysis shows that the initiating E2 adds short K11 containing chains. The second E2 employed by the APC, Ubc1, has been shown to be K48 specific. This suggests that, while there is an interesting intersection between these systems, the human and yeast APC use K11 in distinct manners. It is important to remember that, while K11 linked chains are important for mammalian APC biology, the E2 that generates them, Ube2S, is not essential in human cells. Similarly, several of the APC subunits themselves can be deleted in yeast, and likely human cells. This does not mean that these proteins are not important, but shows that there is some buffering capacity in the cell for this critical enzyme.

Note: We now include the *gnp1∆* data in the supplemental figures.

Additional points:1) It would be informative if the authors were to show FACS profiles for the analyses described in Figure 4.

We now include FACS cell cycle profiles in Figure 4—figure supplement 1. Importantly, the SK1 strain background is strongly flocculent, making FACS analysis difficult to interpret.

2) The result of the Figure 4B would be more compelling if the authors were to include quantification of band intensities in Figure 4. Also include additional loading controls.

The quantified band intensities for the experiments in Figure 4 are now reported. These are now presented in graphical form and exact half-lives are calculated.